# Four chromosome scale genomes and a pangenome annotation to accelerate pecan tree breeding

John T. Lovell [1,15✉], Nolan B. Bentley [2,15], Gaurab Bhattarai[3,15], Jerry W. Jenkins [1,15], Avinash Sreedasyam [1,15], Yanina Alarcon [4], Clive Bock[5], Lori Beth Boston[1], Joseph Carlson[6], Kimberly Cervantes[7], Kristen Clermont[8], Sara Duke[9], Nick Krom[4], Keith Kubenka[10], Sujan Mamidi[1], Christopher P. Mattison [8], Maria J. Monteros [4], Cristina Pisani[5], Christopher Plott[1], Shanmugam Rajasekar[11], Hormat Shadgou Rhein[7], Charles Rohla[4], Mingzhou Song[12], Rolston St. Hilaire[13], Shengqiang Shu [6], Lenny Wells[14], Jenell Webber[1], Richard J. Heerema [12], Patricia E. Klein [2], Patrick Conner[14], Xinwang Wang[10], L. J. Grauke [10], Jane Grimwood [1], Jeremy Schmutz [1,6✉] & Jennifer J. Randall[7✉]

Genome-enabled biotechnologies have the potential to accelerate breeding efforts in long-lived perennial crop species. Despite the transformative potential of molecular tools in pecan and other outcrossing tree species, highly heterozygous genomes, significant presence–absence gene content variation, and histories of interspecific hybridization have constrained breeding efforts. To overcome these challenges, here, we present diploid genome assemblies and annotations of four outbred pecan genotypes, including a PacBio HiFi chromosome-scale assembly of both haplotypes of the 'Pawnee' cultivar. Comparative analysis and pan-genome integration reveal substantial and likely adaptive interspecific genomic introgressions, including an over-retained haplotype introgressed from bitternut hickory into pecan breeding pedigrees. Further, by leveraging our pan-genome presence–absence and functional annotation database among genomes and within the two outbred haplotypes of the 'Lakota' genome, we identify candidate genes for pest and pathogen resistance. Combined, these analyses and resources highlight significant progress towards functional and quantitative genomics in highly diverse and outbred crops.

[1] Genome Sequencing Center, HudsonAlpha Institute for Biotechnology, Huntsville, AL, USA. [2] Department of Horticultural Science, Texas A&M University, College Station, TX, USA. [3] Institute of Plant Breeding, Genetics & Genomics, University of Georgia, Athens, GA, USA. [4] Noble Research Institute, Ardmore, OK, USA. [5] USDA Southeastern Fruit and Tree Nut Research Laboratory, Byron, GA, USA. [6] DOE Joint Genome Institute, Berkeley, CA, USA. [7] Department of Entomology, Plant Pathology and Weed Science, New Mexico State University, Las Cruces, NM, USA. [8] USDA-ARS Food Processing and Sensory Quality Research, New Orleans, LA, USA. [9] USDA-ARS Plains Area Administrative Office, College Station, TX, USA. [10] USDA Pecan Breeding and Genetics, College Station, TX, USA. [11] Arizona Genomics Institute, University of Arizona, Tucson, AZ, USA. [12] Department of Computer Science, New Mexico State University, Las Cruces, NM, USA. [13] Plant and Environmental Sciences, New Mexico State University, Las Cruces, NM, USA. [14] Department of Horticulture, University of Georgia-Tifton Campus, Tifton, GA, USA. [15] These authors contributed equally: John T. Lovell, Nolan B. Bentley, Gaurab Bhattarai, Jerry W. Jenkins, Avinash Sreedasyam. ✉email: jlovell@hudsonalpha.org; jschmutz@hudsonalpha.org; jrandall@nmsu.edu

While modern breeding has produced significant evolutionary bottlenecks in most major crops[1,2], genetic diversity of many other economically and culturally important specialty crop species remains largely untouched. This is especially true for newly emerging, orphan, and long-lived perennial crops, which are often not amenable to accelerated breeding regimes[3,4]. The broad genetic diversity available in specialty crops will be crucial when adapting cultivars to new or changing pests, environmental conditions, and consumer demands.

Pecan (*Carya illinoinensis*) is one such specialty crop. First transported outside its endemic range by Native Americans[5], pecan is now cultivated on six continents[6]. While newly worldwide cropping will undoubtedly expose the species to a number of novel diseases and pests, pecan has co-evolved with many pests and pathogens in its endemic range; these include multiple species of *Phylloxera* (a genus of gall-forming aphid-like insects[7]) and other insect species that can significantly reduce yield[6,8–10], and a genetically diverse phytopathogenic fungus (*Venturia effusa*) that causes scab disease[11,12], which is the most economically damaging disease of pecan[6,10,13]. Despite a paucity of information on the cellular and genetic mechanisms responsible for susceptibility to these pests and pathogens, several resistant cultivars have been bred to mitigate some yield losses[6,10].

Compared to the dramatic morphological evolution during domestication of many major crops[12], modern pecan breeding efforts have thus far resulted in only modest improvements. For example, pecan nuts collected from prehistoric Native American archeological sites appear very similar to present-day cultivars[5]. This is due, in part, to the fact that traditional breeding efforts in pecan and other tree crops can take many decades. Consequently, the primary stocks used in contemporary pecan breeding were derived from crosses made from wild trees during the early twentieth century[6]. Nonetheless, modern pecan breeding has made some significant strides by selecting for genotypes with larger nut size[14], higher nut quality, and tree tolerances of abiotic and biotic stresses[15]. Thus, the development of molecular markers for agronomic traits, which can be assayed early in life, will dramatically improve the speed, efficiency, and efficacy of selection[3] in long-lived perennial crops such as pecan.

Beyond their long lifespan, the outbred and highly diverse nature of pecan and many other tree crops can also complicate molecular breeding goals. For example, breeding programs in pecan and other tree species commonly seek traits originating from highly diverged populations, subspecies, or even related species[16]. Therefore, it is likely that some genes that could be targets for selection are simply not present in many genotypes. Such diversity, both within and among individuals, makes reliance on a single inbred 'reference' genotype untenable and necessitates a paradigm shift towards the use of multiple and outbred genomes.

Here, we construct and analyze four outbred de novo pecan genome assemblies and annotations as a step towards identifying candidate genes and molecular targets for accelerated breeding efforts in outbred and diverse crops. Our efforts to define gene presence–absence variation through a pan-genome annotation reveal evidence of widespread interspecific genomic introgressions. These introgressions and extensive gene content variation between meiotically homologous chromosomes provide a wealth of nut quality and biotic stress resistance genetic diversity that breeders can leverage to improve contemporary and future pecan nut production.

## Results

### Four pecan genomes provide a crucial resource for crop improvement

The outbred nature of the pecan genome[17] complicates genome assembly methods and efforts to leverage genome resources for breeding goals. For example, since genetic mapping in outbred perennial species typically uses $F_1$ breeding designs[18], causal variants may segregate within pecan genomes. Therefore, it is crucial to generate genome assemblies of both meiotically homologous haplotypes in outbred diploids. Nonetheless, past genome sequencing efforts in outbred species have typically sought to represent a single haploid assembly for a genotype either via sequencing an inbred 'reference' genotype (e.g., B73 maize) or phasing a highly diverged $F_1$ hybrid assembly. However, pecan inbred pedigrees are neither biologically realistic (high inbreeding depression) nor practically feasible (long generation times).

Recent assembly methodological improvements and sequencing technology have permitted highly contiguous genome assemblies of switchgrass[19] and several other outbred plant genotypes. Here, we have expanded upon these efforts—instead of collapsing two divergent haplotypes into a single haploid assembly, we sought to build diploid assemblies and capture both haplotypes in four outbred pecan genotypes. To this end, we selected four genotypes to represent the genetic diversity of pecan ('Pawnee'[20], 'Lakota'[21], 'Elliott'[22], and a wild collection from Oaxaca, Mexico '87MX3-2.11', hereon 'Oaxaca'[23]; see 'Methods' section and Supplementary Fig. 1 for details).

Three genomes ('Oaxaca', 'Elliott', and 'Lakota') were assembled using a whole-genome shotgun sequencing strategy combining PacBio single-molecule real-time (RS II, and SEQUEL I, 55.2–108.3 Gb raw long reads, 78.9×–135.3× coverage) and Illumina (HiSeq X Ten and 2500, 50–60× coverage) technologies (Fig. 1a, Table 1). Contigs along the primary haploid path were oriented, ordered, and joined together into 16 highly contiguous ($N_{50} = 3.7$–4.4 Mb, Table 1) chromosome pseudomolecules using synteny and Hi-C indexed short-read sequencing. The remaining alternative haplotype contigs ($n = 3,705$–6,853, $N_{50} = 0.10$–0.14 Mb) represented the homologous sequence to the primary path. In total, the alternative haplotypes captured 64.5–76.1% of the total genomic sequence (Table 1). The missing sequence and relatively short contigs resulted from runs of homozygosity that are expected in breeding pedigrees and highly repetitive pericentromeric regions.

In contrast to these three genomes, the 'Pawnee' assembly was built with state-of-the-art PacBio circular consensus sequencing reads ('CCS' a.k.a. 'HiFi', mean coverage = 52.1×), 'Pawnee' is by far the most contiguous of the four assemblies (contig $N_{50} = 26.5$ Mb) with 100% of primary path sequence assembled into chromosomes (Table 1). Crucially, the highly accurate CCS reads permitted the construction of haplotype-aware contigs even in homozygous and repetitive regions. Nearly 90% of the primary 'Pawnee' assembly size was captured in the highly contiguous (contig $N_{50} = 2.9$ Mb) chromosome-scale alternative haplotype assembly (Table 1). We independently annotated each pecan genome assembly through homology and RNA-seq supported methods (Fig. 1a, Supplementary Table 1), which produced very complete annotations (BUSCO scores 94.4–97%). We leveraged these protein-coding DNA sequences to validate the accuracy and completeness of our assemblies.

The Juglandaceae family (walnut, hickory, pecan) experienced a whole-genome duplication (WGD) event ca. 60 million years (Myr) before present[24], resulting in pairs of homeologous chromosomes with highly conserved paralogous gene order collinearity (i.e., synteny). Reanalysis of syntenic orthologous and paralogous gene blocks in the recently published walnut genome[25] and our 'Pawnee' assembly revealed a total of 26 large homeologous collinear gene blocks in walnut and 25 such blocks in pecan across 16 chromosomes. This represented an exceptional level of chromosomal evolutionary conservation (one

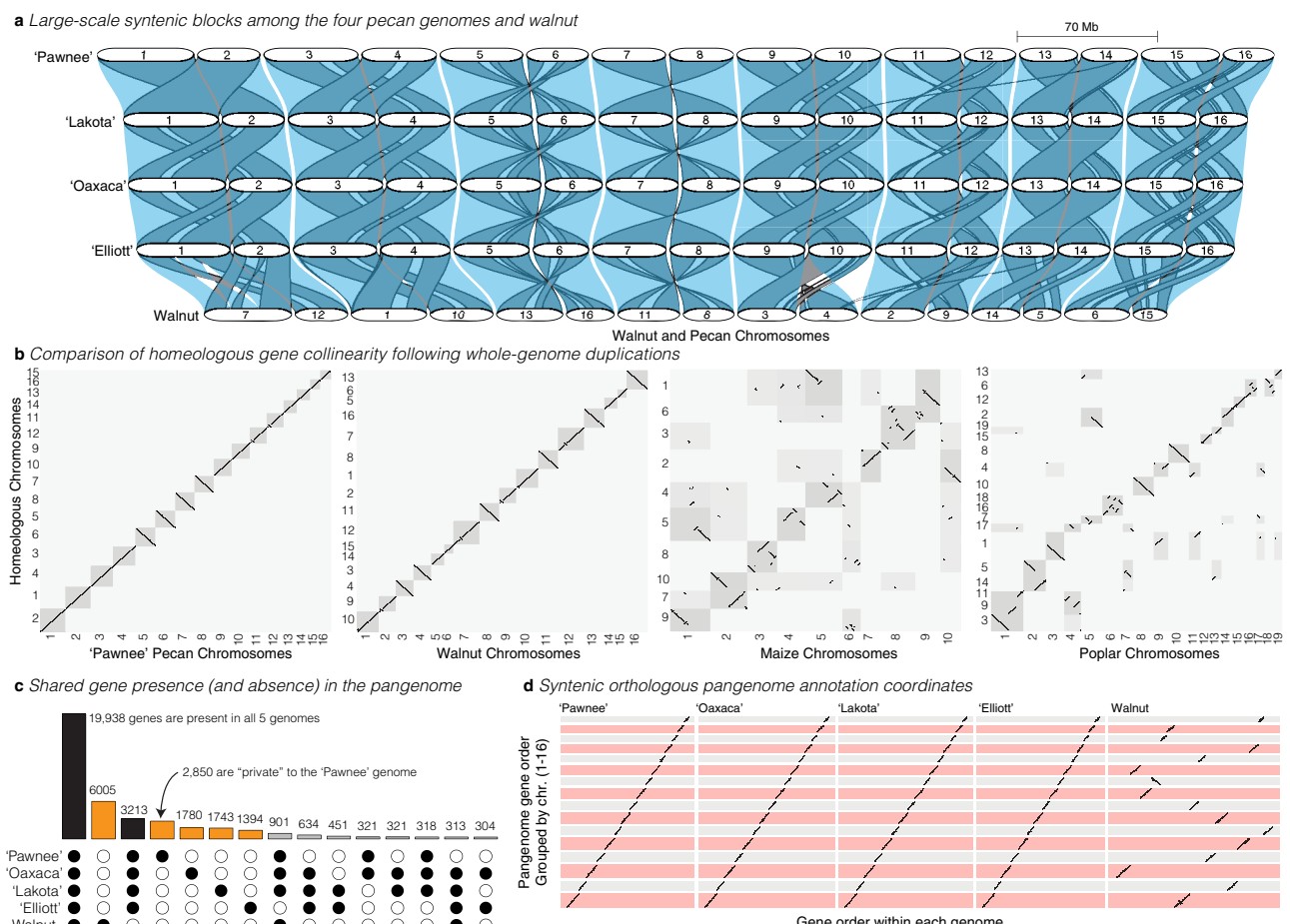

**Fig. 1 Comparative analysis of four de novo pecan genomes. a** A map of syntenic orthologous (transparent blue) and homeologous blocks (gray with black borders) among the four reference genomes and the walnut outgroup. Chromosomes are represented by white segments and are scaled to the same physical size (Mb: megabases) for all genomes. Orthologous chromosomes are stacked vertically and labeled accordingly. **b** Comparisons of the degree of synteny between homeologous chromosomes across the 'Pawnee', walnut, maize, and poplar genomes. The dotplots display the gene-rank-order positions of syntenic blastp hits along the main genome (x axis) and homoeologous chromosomes (y axis). Chromosomal bounds are shaded by the total number of blast hits found between each pair of homeologous chromosomes. **c** Across the pan-genome, the vast majority of all genes are found in orthogroups that contain all four pecan genomes (bars shaded black); however, genes private to each genome (shaded orange) and, to a lesser degree, shared among >1 genome (gray) are also common. Filled circles represent presences in orthogroups; open circles are absences. **d** The high level of synteny between the pecan genomes and walnut allowed for simple pan-genome construction and gene ordering. Here, each point represents the location of a gene by its rank-order location within each de novo genome assembly (x axis) and the inferred syntenic position in the pan-genome (y axis). Source data are provided as a Source Data file.

rearrangement every 6.7 Myr in pecan, Fig. 1b). For context, comparative genomics using the same parameters revealed one rearrangement between homeologs every 800,000 years in poplar (WGD ~58 Myr ago[26,27]) and 490,000 years in maize (WGD ~12 Myr ago[28,29]; Fig. 1b). While shorter generation time in the progenitors of maize, and possibly poplar, certainly could have contributed to elevated chromosomal evolutionary rates, paralogous synteny in pecan represents a remarkable level of chromosomal stability over 60 M years. Crucially, such retained synteny offers an opportunity to validate genome completeness and contig ordering by comparing synteny between homeologous chromosomes. Since our pecan genomes were assembled agnostic to homeologous chromosome synteny, this level of conservation lends credence to the assertion that these four genomes were very complete and lack any major assembly errors.

**A pan-genome representation of pecan gene diversity**. Genome evolution and genetic diversity that underlie breeding targets can

arise from a diverse set of genetic and epigenetic changes including short insertions/deletions (INDELs), single nucleotide polymorphisms (SNPs), structural variants, and presence–absence variants (PAVs). Our four de novo genome assemblies and annotations permit the inference of each of these variant types through comparative analysis of a database of conserved and variable orthologous gene sequences among all genomes, a 'pan-genome annotation' (Fig. 1c, d). In clades with a history of whole-genome duplications such as Juglandaceae, pan-genome construction methods based solely on sequence homology are not sufficient for comparative genomics since paralogous sequences would likely pollute otherwise orthologous gene families. For example, 16.2% of genome-wide 'Pawnee' orthogroups contained homeologous gene pairs. To overcome this genome complexity, we constructed a synteny constrained orthologous pan-genome annotation (Supplementary Data 1 and Fig. 1), which simultaneously masked paralogous regions and condensed tandem arrays into a single orthologous path through multiple genomes. While offering a powerful method to reduce paralogous gene content in the pangenome, it is important to note that constraining to

**Table 1 Genome assembly and annotation statistics for each of the four genomes.**

| Genomic features | 'Oaxaca' | 'Lakota' | 'Elliott' | 'Pawnee' |
|---|---|---|---|---|
| Assembly size (Mb)[a] | 649.96 | 668.99 | 656.69 | 674.27 |
| Number of scaffolds | 298 | 261 | 431 | 16 |
| Number of contigs | 552 | 499 | 829 | 34 |
| Gap content (%) | 0.4% | 0.4% | 0.6% | 0.0% |
| Contig N50 (Mb) | 4.4 | 3.7 | 4.4 | 26.5 |
| Genome in chromosomes (%) | 98% | 96.1% | 95.5% | 100% |
| Number of annotated genes | 31,911 | 33,280 | 31,042 | 32,267 |
| Average number of exons per gene | 5.4 | 5.5 | 5.5 | 5.5 |
| Repeat sequences (%) | 46.5% | 33.8% | 32.3% | 49.7% |
| Total alt. haplotype size (Mb)[b] | 494.9 | 469.7 | 423.6 | 603.2 |
| Number of alt. haplotype scaffolds | 6,853 | 5,222 | 3,702 | 16 |
| Number of alt. haplotype contigs | 6,853 | 5,222 | 3,702 | 323 |
| Alt. haplotype contig N50 (Mb) | 0.13 | 0.10 | 0.14 | 2.90 |
| Alt. genome size (% of main) | 76.1% | 70.2% | 64.5% | 89.5% |

[a]Statistics extracted for the primary ('main', top section) assembly.
[b]Alternative haplotype (alt.) are presented in the bottom five rows.

syntenic regions will ignore orthologs involved in very small chromosomal translocations. Overall, these minor translocations represent <0.4% of the genome.

Rooted against gene order of the 'Pawnee' genome and including walnut (*Juglans regia*)[25] as an outgroup, the pan-genome annotation contained 42,416 orthogroups, 21,196 of which were single-copy in all four pecan genomes (Fig. 1c). Among the four pecan genomes, the synonymous ($K_s$) and non-synonymous ($K_a$) nucleotide substitution rates of single-copy syntenic orthologs were fairly low (mean $K_a \pm SEM = 0.0017 \pm 1.24 \times 10^{-5}$; $K_s = 0.0042 \pm 2.4 \times 10^{-5}$; Supplementary Fig. 2 and Fig. 2a). This evolutionary conservation was exemplified by allergen proteins, which tend to be highly conserved in walnut and other tree nut crops[30]. Pecan allergic reactions are caused by immunoglobulin-E (*IgE*) recognition and binding of Car i 1, 2, and 4 allergen protein structures[31]. Like many of the orthogroups present in all four genomes, coding sequences of the Car i 1 and Car i 2 allergens were nearly identical between genotypes and IgE binding epitopes were conserved (Supplementary Fig. 3). We observed only a single amino acid substitution in the 'Oaxaca' Car i 1 and 'Elliott' Car i 2 alleles respectively. However, some unique differences were observed in the Car i 4 sequences among the cultivars, where 'Elliott', 'Oaxaca', and 'Lakota' shared a 33 amino acid exon, which may have differentiated the allergen profile of 'Pawnee'.

In contrast to the constrained coding sequence evolution of single-copy genes, we observed significant gene PAV among these relatively closely related genomes. Overall, 38.7% of orthogroups in the pan-genome ($n = 13,010$) were incomplete, representing PAV among the pecan genomes (Fig. 1c and Supplementary Data 1). To dissect the differential roles of gene-model structural evolution, sequence deletion, and evidence-based gene model thresholding on PAV, we compared sequence similarity between genes present in one annotation and the syntenic unannotated genomic regions where absent genes should exist (Supplementary Table 2). Overall, a majority of the observed pan-genome PAV was driven by gene sequences that were unannotated yet similar to annotated sequences in alternative genomes. As observed previously[32], such genes tended to be of low-quality barely passing gene evidence score thresholds. However, 8,655 absent genes had no similar sequence within syntenic regions, indicating significant and diverse mechanisms of gene absence among our genomes.

Among the four pecan genomes, we observed 3,889 blocks of five or more consecutive genes that were absent in one or more of the references. Many of these gaps represented true gene absences and demonstrated that multiple reference genomes offered a major improvement in gene content representation over a single pecan genome. Further, the ubiquity of large runs of genes that were unique to a single genotype ('private' genes) potentiated a role for independent genomic introgressions from distantly related gene pools into each of our reference genome lineages, a hypothesis we test below.

**Genomic introgressions as breeding targets for disease resistance.** In addition to conspicuous runs of PAV within each genome, we observed several physical regions of elevated divergence among the four genomes (Supplementary Fig. 2 and Fig. 2a). While a number of factors could cause these divergence peaks, ancient and contemporary hybridization and admixture offer one potential reason for the observed high level of PAVs, long runs of private genes, and regions of elevated nucleotide substitutions in each assembly. Indeed, there are records of historical pecan breeding incorporating progeny from bitternut (*C. cordiformis*) and other interspecific pedigrees[16]. Furthermore, morphological analysis of extant trees[33] and remains from pre-historical archeological sites[5,34,35] in Mexico found a strong affinity to *C. aquatica* and *C. myristiciformis*, indicating that ancient admixture between *Carya* species may have imbued pecan with desirable traits for human cultivation.

Given the complete sequence order of our assemblies, it was possible to track the positions and identities of genomic introgression blocks from these three related species into pecan breeding pedigrees. To estimate introgression proportions and positions, we resequenced multiple genotypes of each of these three potential admixing species (*C. cordiformis*, *C. aquatica* and *C. myristiciformis*) and defined admixture blocks by decoding SNP-based (38–69× coverage of Illumina 2 × 150 bp reads) posterior ancestry probabilities among the four reference genomes and three or four relatives of each reference genotype (Fig. 2b and Supplementary Table 3). By including multiple relatives for each genome, we were able to define high-confidence interspecific genomic introgressions as regions with non-pecan ancestry in all related genotypes. While these introgressions represented sequences from other species, it is important to note that some introgressions may have been derived from other unsampled species.

Overall, *C. aquatica* (or an unsampled related species) was the primary source of interspecific introgressions, representing 6.6–20.6 Mb (1.04–3.23%, Supplementary Table 3) of the entire genome sequence of the four reference genomes. These introgressions tended to be small and distributed regularly across the genome (Fig. 2b), indicating that *C. aquatica*'s hybridization history may have begun long before modern pecan breeding efforts. This seems particularly plausible given the largely sympatric geographic distributions of the two species. The physically discontinuous, yet high levels of *C. aquatica* ancestry likely contributed to the significantly elevated synonymous substitution rates on chromosome 5 (right arm), 14 (right arm), and 16 (Supplementary Fig. 2 and Fig. 2a).

In contrast to a putatively ancient and natural origin of admixture between pecan and *C. aquatica*, the vast majority of *C. cordiformis* ancestry was concentrated in a >7.5 Mb block on chromosome 8 derived from the 'Major'[16] cultivar and present

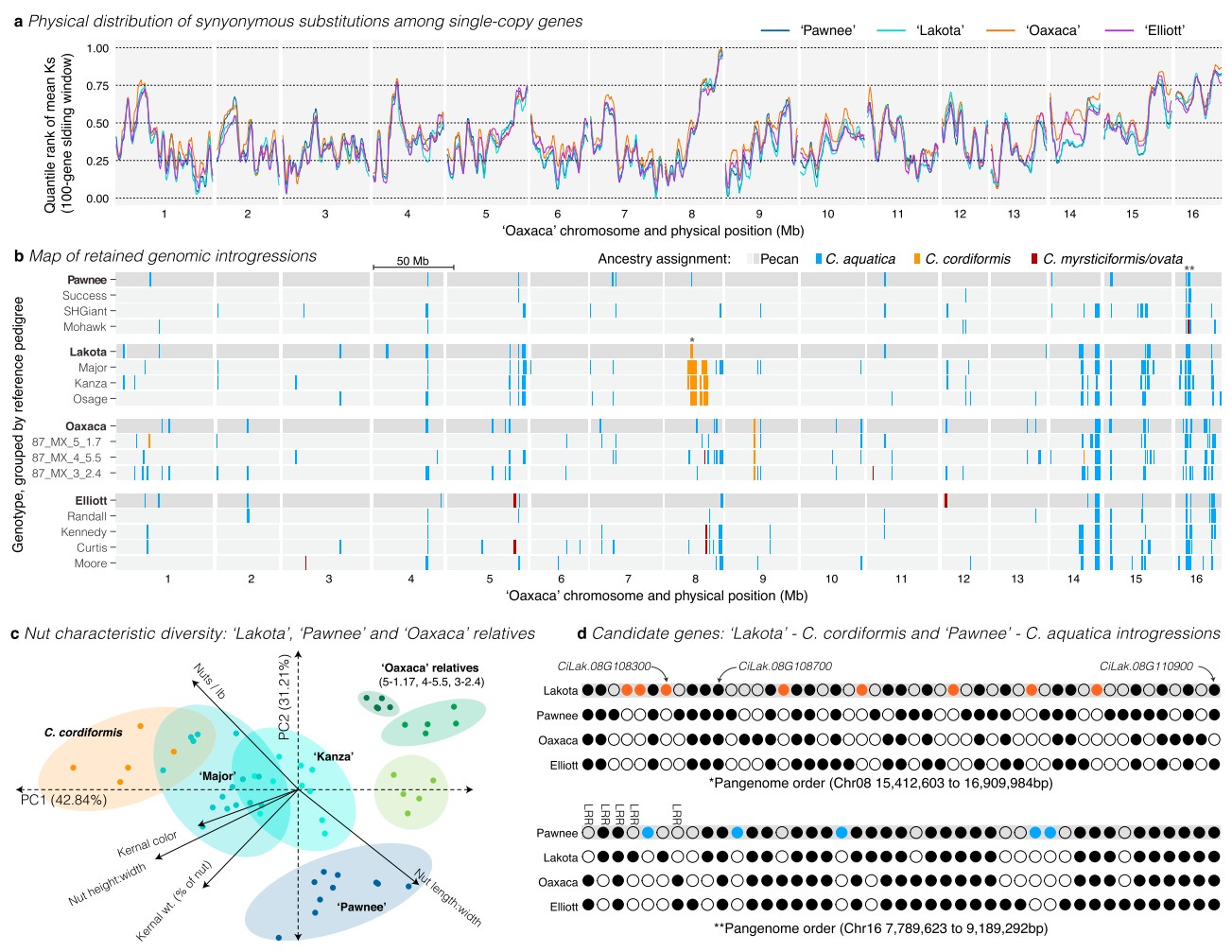

**Fig. 2 A map of interspecific genomic introgressions in four pecan genomes. a** Sliding window analysis of neutral site substitution rate ($K_s$) within all single-copy orthogroups that were represented by all four genomes. $K_s$ values were transformed to quantiles and a 100-gene sliding window was applied within each chromosome and genome. The resulting sliding window values are presented on a 0–1 scale where lower values represent the most similar regions across the physical genome (Mb: megabases). See Supplementary Fig. 2 for raw pairwise $K_s$ values. Close-up pan-genome representations of two regions marked * and ** are highlighted in **d**. **b** Genome ancestry maps of the four reference genomes and representative members of each pedigree. Posterior probabilities of ancestry for three primary hybridizing species were decoded into blocks (colors red, orange, blue) of ≥500 variants. The background pecan ancestry is dark and light gray for the reference genomes and relatives respectively. **c** The large introgression in the 'Major' and 'Kanza' relatives of 'Lakota' appear to imbue phenotypic variation typical of *C. cordiformis* to these genotypes. 13 traits associated with nut yield and quality were assayed for a single *C. cordiformis* genotype (02-COR-LA-BF1), 'Pawnee', two members of the 'Lakota' pedigree ('Major' and 'Kanza') and three genotypes from Mexico that may be related to 'Oaxaca'. The 13 traits were reduced to five non-collinear ($|r| < 0.75$) representatives and decomposed into the two major principal component axes (PC1, PC2), which collectively explained >74% of the variation. For each genotype, we present the positions in PCA space and the 95% confidence ellipse. **d** Pan-genome gene representatives are shown for each unique orthogroup within two physical (base pairs, bp) introgression intervals. Circles represent presence (filled) or absence (open) for each genome (row) by orthogroup (column) in the introgression. The first row in each plot represents the genome into which an introgression was observed. Private orthogroups to that genome are colored following panel **b**. Three candidate genes in 'Lakota' and the dense region of leucine-rich repeat (LRR) genes are annotated along the top row of each map. Source data underlying Fig. 3a–c are provided as a Source Data file. Raw data associated with **d** can be found within the pangenome database in Supplementary Data 1.

among four genotypes related to 'Major' (Table 1, Fig. 2b; 'Lakota' ['Mahan' × 'Major'], 'Kanza' ['Major' × 'Shoshoni'], and 'Osage' ['Major' × 'Evers']). The presence of a single large introgression indicated (1) a recent origin, (2) positive selection to retain the introgressed region, and (3) purifying selection against all other *C. cordiformis* haplotypes across the genome. Additionally, while 'Lakota', 'Major', 'Kanza', and 'Osage' have overall pecan-like morphologies they possess other traits that cluster closely with *C. cordiformis* (Fig. 2c). Indeed, 'Major' (and to a lesser degree 'Kanza') also had the most similar nut characteristics to *C. cordiformis* of a subset of genotypes related to 'Pawnee', 'Lakota' and 'Oaxaca' (Fig. 2c).

'Lakota', 'Major' and other members of this pedigree are also known to have strong fungal and abiotic stress resistance[16], traits that could be due to shared ancestry across the large introgression on chromosome 8. To explore this hypothesis, we examined the *C. cordiformis* introgression interval in 'Lakota', which was much narrower than the intervals in other members of its pedigree. Such introgression size reduction in a single generation indicated that recombinant gametes at the margins of this introgression were selected by breeders. The 1.41 Mb region contained 24 high-value candidate genes in the 'Lakota' genome (Supplementary Data 2), many of which had homologs in other species known to be involved in nutrient acquisition, plant development, and

defense responses including *SNF1*-related protein kinase and Leucine-Rich Repeat (LRR) receptors. The pan-genome database of this region contained 46 total orthogroups, eight of which were private only to 'Lakota' (Fig. 2d). The 17.4% of private genes unique to 'Lakota' represented a >4-fold enrichment in private gene content compared to the genome-wide average (Fisher's exact test, odds ratio = 4.232, $P = 0.0012$), demonstrating evidence of non-pecan ancestry from both SNPs and PAV datasets.

In addition to the chromosome 8 *C. cordiformis* introgression, there were a number of other high-confidence introgressions that appeared in multiple related genotypes (Fig. 2b and Supplementary Fig. 2). For example, chromosome 5 and 16 harbored introgressions from *C. myristiciformis* into 'Elliott' and *C. aquatica* into the 'Pawnee' pedigree, respectively. For each of these regions, we queried the pan-genome and extracted the synteny constrained orthogroups within each focal genome annotation (Supplementary Data 2). The introgressed region on chromosome 5 was characterized by plant signaling genes (there are no less than 10 cell wall receptor kinases) and cell wall defense genes including lignin biosynthesis genes (4 genes), cellulose synthase, and inositol oxygenase, which involved in cell wall polymerization. The region of *C. aquatica* introgression into 'Pawnee' on chromosome 16 contained nine LLR receptor serine/ threonine kinase genes from five unique orthogroups (Fig. 2d). The apparent overabundance of defense-related genes within introgression regions hints at a possible adaptive role for introgressions in both pecan breeding and wild populations.

**Induced gene networks in a pathogen susceptible cultivar.** Biotic stress tolerance is a major breeding objective in many crops, but especially in long-lived tree species where pests and disease incidence varies across years and locations[10,36]. A temporally and spatially variable pathogen composition can obfuscate breeding values, and subsequently, reduce the efficacy of traditional breeding efforts. Given these constraints, generating molecular targets for resistance to specific pathogens can dramatically accelerate crop improvement outcomes[37–39]. For example, pecan scab (caused by the phytopathogenic fungus *V. effusa*) produces black circular lesions that can reduce yield and nut quality, and if not controlled, can cause crop failure[40]. *V. effusa* is composed of multiple pathotypes each capable of infecting a relatively small subset of pecan cultivars[41]. Most benign *V. effusa*-pecan cultivar interactions result in the arrest of fungal growth shortly after cuticular penetration, whereas virulent interactions result in abundant intercellular hyphal growth and sporulation[42]. Natural populations of pecan present the host with a diverse and evolving host, limiting the buildup of virulent races. In contrast, pecan orchards composed of replicated stands of only a few cultivars promote the accumulation of pathogenic strains[41,42]. In recent years several major industrial pecan cultivars, including the most widely planted cultivar in the southeastern U.S. ('Desirable'), have become more susceptible to scab infection[10].

To understand susceptibility in 'Desirable' and the landscape of short-term gene-expression plasticity to *V. effusa*, we compared transcript abundance in leaf tissue inoculated with the scab isolate 'De-Tif-11' compared to the control treatment (Supplementary Fig. 4) through sequencing of RNA extracted across three biological replicates at 24 h post inoculation (Supplementary Table 4). While we did not generate a genome assembly and annotation for 'Desirable', the phylogenetic dispersion of the four pecan genomes covers much of the pecan diversity whereby 'Pawnee' and 'Desirable' share a grandparent. Of the 32,267 genes in the 'Pawnee' reference, 194 genes were differentially expressed ($|\text{Log}_2$ fold-change$| \geq 1.5$ and FDR-adjusted $P$-value < 0.05)

between control and inoculated tissue, showing strong evidence of molecular phenotypic plasticity to the fungal pathogen treatments (Supplementary Fig. 4 and Supplementary Data 3). While gene ontology (GO) term enrichments from such differential expression analysis can be vague and imprecise, GO enrichments in this experiment were clear (Supplementary Table 5): by far the most significant terms were 'response to wounding' (downregulated genes) and 'response to chitin' (upregulated genes). Other significantly enriched terms were heavily biased towards stress responses and oxidation–reduction status. Since chitin is the primary trigger of plant responses to fungus[43,44], and redox status is crucial to plant defense responses[45,46], these differentially expressed genes offer a set of targets to explore host susceptibility to *V. effusa*.

**PAV within genomes to target candidate genes in outbred pedigrees.** While genome-informed molecular and genetic diversity exploration can document potentially important breeding targets, these efforts lack causality at a per-locus level; however, linkage-based quantitative trait locus (QTL) mapping can identify sequences in linkage disequilibrium with causal variants. Due to pecan's long generation times and inbreeding depression, breeding programs have utilized pseudo-testcross ($F_1$) mapping strategies[47–49] to identify causal variants that segregate within the genomes of one or both parents. We applied this mapping strategy to test the genetic basis of phylloxera leaf gall incidence caused by feeding of the larva of aphid-like phylloxera insects (order Hemiptera) among 143 2-year old 'Lakota' × 'Oaxaca' $F_1$ saplings (Supplementary Data 4) at a nursery in Somerville, TX. Linkage phases of 11,489 loci (Supplementary Data 5) that were heterozygous in 'Lakota' and homozygous in 'Oaxaca' were defined by the parent of origin ('Mahan' or 'Major') based on comparison of the phased marker positions in common with a previous restriction-enzyme site associated sequencing of 'Mahan' and 'Major'[48,49].

QTL mapping revealed a single large peak on chromosome 16 (Fig. 3a and Supplementary Data 5). Given a left-skewed phenotypic distribution, the peak logarithm of the odds (LOD) score of 14.8 approached the maximum possible value in this experimental design. Indeed, all but two of the individuals with the 'Mahan' haplotype at the peak QTL position (2.021 Mb, Fig. 3a) were completely free of phylloxera galls, while all highly susceptible genotypes inherited the 'Major' haplotype (Fig. 3b).

To define the candidate genes that may explain phylloxera gall incidence in this population, we explored the syntenic pan-genome region in 'Lakota' that corresponded to the 95% Bayes credible QTL interval from positions 1,481,054–2,615,216 bp in the chromosome 16 sequence in the 'Oaxaca' assembly (Fig. 3c and Supplementary Table 6). Since causal variants in $F_1$ experimental crosses are heterozygous within the parents, we ranked candidate genes by the level of divergence between the primary and alternative haplotype of the outbred 'Lakota' genome assembly. Overall, the 'Lakota' genomic interval contained 40 genes only found in the primary assembly. The bulk of these genes resided within a block where no homologous alternative sequence was assembled. These likely represent homozygous regions that were less likely to contribute to the QTL. However, 22 gene models were only found in the alternative assembly (Fig. 3c, unfilled circles in alternate sequence) and represented higher-confidence PAV since both haplotypes across these loci were assembled but genes were only annotated on one sequence. Finally, 12 genes were found in both assemblies but with peptide identities of <98% between haplotypes. Based on this logic, the aforementioned 22 and 12 gene model groupings were made high-priority candidate genes.

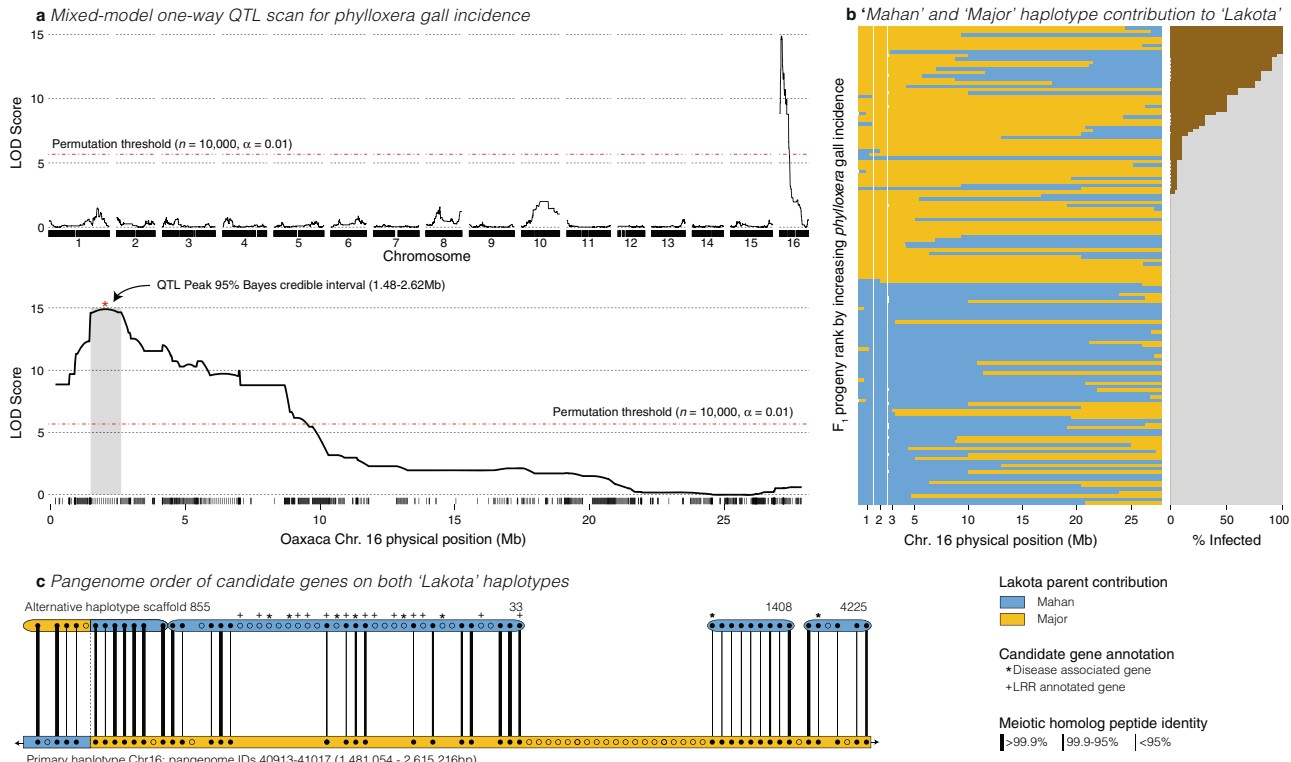

**Fig. 3 Analysis of a major QTL for phylloxera resistance. a** Quantitative Trait Locus (QTL) scans, controlling for genomic background via the leave-one-chromosome-out method for % phylloxera gall incidence. This experiment was conducted once at a single time point. Since the phenotype is non-normal, we determine the significance of QTL peaks via 10,000 permutations. The full genome and a close-up visualization of chromosome 16 are presented along the physical position (Mb: megabases) of the 'Oaxaca' genome assembly. The 95% confidence interval surrounding the QTL peak is shaded. **b** As evidenced by very high LOD scores for a 140-genotype population, there is an extremely strong haplotype structure at the peak QTL (between the vertical white bars), where all but two individuals that inherited the 'Mahan' haplotype from 'Lakota' have no evidence of phylloxera galls (gray horizontal bars in the plot to the right), while all individuals with >50% phylloxera gall incidence retained the 'Major' haplotype at the QTL peak region (brown horizontal bars indicate % incidence). **c** To define candidate genes, we queried the pan-genome within the physical bounds (base pairs, bp) of the QTL interval. All unique genes in this interval were projected onto the alternative haplotype; those contigs where >50% of the projected genes were derived from the candidate interval were extracted and aligned to the primary haplotype. Orthologous genes between the two haplotypes are connected by a solid line, the thickness of which is scaled by % identity between the two protein sequences. Presence–absence variant (PAV) genes without a projected ortholog are represented by open circles. Homologs of the genes in the interval were queried in model systems and qualified by whether annotations indicated a disease-related function or a leucine-rich repeat (LRR) motif. Finally, the haplotypes were coded by whether they were derived from the 'Mahan' or 'Major' parents of 'Lakota'. Source data underlying **c** are provided as a Source Data file. Raw data associated with **a**, **b** can be found in Supplementary Data 5.

Analysis of the functional domains (Pfam and PANTHER) and UniProt descriptions of the high-priority candidate genes identified 20 putative plant immune response genes. The largest grouping of these included a series of 13 LRR motif-related genes that were present at variable copy numbers in all the reference sequences but were least numerous in the primary 'Lakota' sequence which represented the 'Major' derived haplotype across this interval. Consistent with LRR-induced phylloxera resistance, studies in other systems have mapped aphid resistance to LRRs in *Capsicum baccatum*[50] and cucumber[51]. While this analysis does not define a single candidate gene, it does inform future efforts to characterize the mechanism of phylloxera susceptibility in pecan by prioritizing experiments of differential LRR-gene-mediated phylloxera gall resistance. Additionally, these candidate variants between haplotypes of the outbred 'Lakota' genome provide target loci for marker-assisted selection to improve phylloxera resistance in several major breeding pedigrees of pecan.

## Discussion

Traditional tree breeding strategies require observation of production-associated traits; yet in species with long juvenile growth periods like pecan, these data can take years to observe. Conversely, marker-assisted, and genomic selection can be implemented prior to plant maturity, dramatically improving the speed and efficacy of selection in long-lived perennial species breeding programs.

The majority of genome-enabled breeding efforts rely on mapping short sequences against a single haploid reference genome. However, outside of a handful of modern domesticated crops, many plant species (including pecan and other tree crops) are characterized by tremendous genetic diversity and often have a history of interspecific hybridization or polyploidy. In these cases, high-value breeding targets may not be easily verifiable when contrasted to a single reference. For example, the candidate genes for the phylloxera susceptibility QTL described here represented a complex genomic region with tandem arrays and extensive presence–absence variation within a single parental genome. Without additional genomic assemblies, annotations and comparisons, mapping short reads to a reference would not have been sufficient to determine the extent of molecular evolution in this region.

Multiple genome assemblies have been recently constructed for many species, revealing a previously undiscovered level of intra- and interspecific genetic exchange[52]. It is becoming clear that

such large-effect evolutionary events are common in nature and may be a source for accelerated selection of high-value breeding targets, especially in emerging and perennial crops. Furthermore, much of the genetic variation in wild and outbred species exists as highly diverged heterozygous haplotypes within individuals. As such, the single-reference-genome paradigm is not sufficient for functional genomics in such systems, not only because heterozygous gene sequence variation cannot be captured by a haploid genome assembly, but also because gene content is highly variable among genotypes. Combined, our approach that integrated comparative and quantitative genomics among multiple outbred de novo genomes shed light into an evolutionary system that would have been poorly represented by a single haploid genome.

## Methods

**Sequenced genotypes and pedigrees.** Four pecan genotypes were selected for complete genome sequencing. 'Oaxaca' was collected in September 1987 as an open-pollinated nut accession from its mother tree near Zaachila, Oaxaca, Mexico[23]. 'Oaxaca', had higher homozygosity than the other three genotypes. As such, and because of a fairly independent pedigree from the other cultivars, we chose to use 'Oaxaca' as the reference to which short reads were mapped for introgression and genetic mapping purposes (see below). The other three are commercial cultivars. 'Elliott' was a seedling of unknown parentage, but possibly descended from trees collected by John Hunt in Mexico during the Mexican war of 1848[22]. While not a heavy nut producer, 'Elliott' produces a relatively small, round nut of very high kernel quality and exhibits a high level of resistance to scab. 'Lakota' was released in 2007 after extensive testing and was derived from the controlled cross of 'Mahan' × 'Major' in 1964[21]. 'Lakota' pecan trees have excellent tree strength, can produce large yields, exhibit early nut maturation, and have excellent scab resistance. 'Pawnee' was the progeny of a controlled cross of 'Mohawk' × 'Starking Hardy Giant' ('SHGiant' in Fig. 2b) in 1963 and released in 1985[20] and is notable for its early nut harvest and excellent resistance to the yellow aphid complex[53]. 'Lakota' and 'Pawnee' share 'Mahan' as an ancestor; 'Mahan' is the mother of 'Lakota' and the maternal grandfather of 'Pawnee'. To place these genotypes in the context of a sample of pecan genetic diversity, we calculated principal components (Supplementary Fig. 1b) derived from the marker-based genetic distance matrix presented in Bentley et al.[48]. The following genotypes were excluded from Bentley et al.'s[48] distance matrix prior to PCA calculation due to either not being a pecan genotype, having low coverage or being a clonal replicate: 'Major', 'Jones Hybrid', 'Abbott Thinshell', '89-XBR-RDM-1', 'Nielson Ovata', '92-AQU-TX-2', '09-CAT-ZH-45', '02-COR-LA-BF-2', 'Ring Palmeri', 'Clark II', 'CloneA-Ramet1', 'CloneA-Ramet2', 'CloneA-Ramet3', 'CloneB-Ramet1', 'CloneB-Ramet2', 'CloneB-Ramet3', 'CloneC-Ramet1', 'CloneC-Ramet2', and 'CloneC-Ramet3'.

**Genome sequencing and assembly.** Leaf tissue was collected from extant trees at the College Station, TX, USA orchard (30.51°N, 96.44°W): CSHQ13-4 ('Pawnee'), CSX8-4 ('Lakota'), CSP1–30 ('Oaxaca') and CSV16-10 ('Elliott'). High molecular weight DNA was extracted for all genomes from young leaves using the protocol of Doyle and Doyle[54] with minor modifications. Size was validated by pulsed-field gel electrophoresis.

The 'Oaxaca' genome was assembled and polished with MECAT v1[55] using 78.94× PACBIO coverage (average read length of 12,163 bp), and the resulting assembly was polished using QUIVER v2.2.2[56]. Misjoins in the assembly were identified using a combination of previously published 774-marker genetic map[49] and HiC scaffolding. A total of nine misjoins were identified in the polished assembly. The main genome consisted of 552 contigs assembled into 298 scaffolds that contained 647.4 Mb of sequence (contig $N_{50}$ = 4.4 Mb, scaffold $N_{50}$ = 42.3 Mb). Scaffolds were oriented, ordered, and joined together into chromosome pseudomolecules using a combination of the Hi-C scaffolds and genetic markers[49]. A total of 298 joins were applied to the broken assembly to form the final assembly consisting of 16 chromosomes, with a total of 97.98% of the assembled sequence contained in the chromosomes. Assembling a diploid genome in an outbred individual requires a computational step to distinguish a primary and alternative haplotype. While we define the primary haplotype as that with the most contiguity, regions that collapse to a single contig due to homozygosity or repeat content may introduce overlapping chromosomal regions that must be represented as a single-copy haplotype without duplicate copies being unnecessarily repeated. To resolve minor overlapping regions on contig ends, adjacent contig ends were aligned to one another using BLAT v36[57] and a total of 44 adjacent alternative haplotypes were identified in the joined contig set and were collapsed using the longest common substring between the two haplotypes. Heterozygous SNPs and INDELs that represented phasing errors were corrected using the 65.01× raw PACBIO data. A total of 119,268 (5.5% of the 2,152,592 heterozygous SNPs/INDELs) were corrected.

'Lakota' and 'Elliott' genomes were assembled in an identical manner to 'Oaxaca', except that syntenic markers with 'Oaxaca' were used to identify misjoins

and joins instead of a de novo genetic map. The syntenic markers consisted of 57,706 1 kb unique, non-repetitive regions extracted from 'Oaxaca' sequences, with a minimum spacing between markers of 20 kb. Assembly and polishing were conducted following the 'Oaxaca' genome, with PacBio coverage (125.01×/135.33× 'Lakota'/'Elliot', respectively; average read length of 11,488/8,835 bp); 61/11 misjoins and 298/4 contig joins were identified with Hi-C and syntenic markers. 60/96 alternative haplotypes were collapsed, and a total of 461,327/56,589 heterozygous SNPs/INDELs phasing errors were corrected with the raw PACBIO data. The Lakota genome contained 669.0 Mb of sequence in scaffolds with a contig and scaffold $N_{50}$ of 3.7 and 41.6 Mb, respectively, and 99.8% of the main genome assembled into scaffolds >50 kb. The Elliot genome contained 652.7 Mb of sequence in scaffolds with a contig and scaffold $N_{50}$ of 4.4 and 41.2 Mb, respectively, and 99.4% of the main genome assembled into scaffolds >50 kb.

The 'Pawnee' main assembly was performed with HifiAsm v0.5[58] using 52.12× CCS coverage (mean read length of 20,869 bp), and the resulting assembly was polished using RACON v0.5[59]. As above, misjoins in the assembly were identified using a combination of 58,192 1 kb unique, non-repetitive syntenic sequences derived from the V1 'Lakota' release, and Hi-C scaffolding using the JUICER v1.5.6[60] pipeline. A single misjoin was identified in the polished assembly. Scaffolds were then oriented, ordered, and joined together using a combination of the Hi-C scaffolds and syntenic markers. A total of 18 joins were applied to the broken assembly to form the final assembly consisting of 16 chromosomes, with a total of 100% of the assembled sequence contained in the chromosomes. Heterozygous SNP/INDEL phasing errors were corrected using the 52.12× CCS data. A total of 559 (0.01% of the 5,428,928 heterozygous SNPs/INDELs) were corrected. Additionally, homozygous SNPs and INDELs were corrected in the release sequence using 50× of Illumina reads (2 × 150, 400 bp insert). The Pawnee primary genome assembly contained 674.3 Mb of sequence in scaffolds with a contig and scaffold $N_{50}$ of 26.5 and 44.7 Mb, respectively. The alternative haplotype genome assembly contained 603.2 Mb of sequence in scaffolds with a contig and scaffold $N_{50}$ of 2.9 and 40.0 Mb respectively.

For all genomes, contigs containing telomeric sequence were identified using the $(TTTAGGG)_n$ repeat, and care was taken to ensure that contigs terminating in this sequence were properly oriented in the production assembly.

**Genome annotation.** Our gene annotation pipeline leveraged both homology and RNA sequencing evidence to build high-confidence gene models. Transcript assemblies were generated from 2 × 150 paired-end Illumina RNA-seq reads using PERTRAN (Lovell et al.[32]; see Supplementary Table 1 for library coverage, read counts, and other metadata). RNA-seq transcript assemblies and ESTs were aligned to the genome assemblies with PASA v2.0.2[61]. Repetitive DNA elements were identified de novo with RepeatModeler v2.0.1[62]. Loci were determined by transcript assembly alignments or EXONERATE v2.4.0[63] alignments of proteins from *Arabidopsis thaliana*[64], *Populus trichocarpa*[27], soybean[65], *Oryza sativa* (var Kitaake)[66], *Sorghum bicolor*[67], *Setaria viridis*[68] and Swiss-Prot[69] proteins to repeat soft-masked genomes using RepeatMasker v4.1.0[70]. Alignment extensions of up to 2,000 bp were permitted on both strands unless the extension overlapped with another locus on the same strand. Homology-based gene model prediction was accomplished via FGENESH v3.1.1/ FGENESH_EST v2.6[71], and GenomeScan v1.0[72]. EST and protein support scores, and down-weighting by overlaps with repetitive regions, were used to determine and select the highest-scoring predictions for each locus. PASA was subsequently used to improve gene models by adding UTRS, splice junctions, and alternative transcripts. The transcripts were selected if its Cscore (homology and coverage weighted gene model score) and protein coverage were ≥0.5, unless >20% of the CDS overlapped with repeats, in which case the Cscore threshold was increased to ≥0.9 and homology coverage to >70%. Finally, gene models with protein were annotated with >30% PFAM TE domains were removed.

**Comparative genomics.** To infer paralogous collinear blocks, we ran orthofinder v2.3.11[73] on pairwise diamond v0.9.36[74] blast-like hits pruned to the top two-bit score hits per gene for each pairwise combination of pecan ('Pawnee' v1.1), English walnut (Chandler v2.0[25]), maize (RefGen v4[28]) and poplar (*Populus trichocarpa* v3.1[27]). The self-blast hits were pruned to cases where both query and target genes were members of the same orthogroups, then to synteny via MCScanX[75] (−m = 50, −s = 10) and dbscan v1.1–5[76] (radius = 50, min. hits = 10). The number of homeologous collinear blocks were determined as the number of MCScanX breakpoints for each non-redundant combination of off-diagonal (not self-hit chromosomes) chromosome pairs and corrected by the base number of chromosomes in each comparison.

We built the pan-genome annotation using GENESPACE[32]. In short, GENESPACE accomplishes synteny constrained orthology inference across multiples species permitting variable ploidy by parsing protein similarity scores into syntenic blocks and runs orthofinder[73] on synteny constrained blast results. The resulting block coordinates and syntenic orthogroups give high-confidence anchors for evolutionary inference. The five-genome pan-genome annotation (with *J. regia* v2.1[25] as the outgroup) was constructed using default settings (minimum block size (b) = 10, radius / gaps (g) = 20, n. hits / gene / haploid genome = 1). Each orthogroup in the pan-genome representation was a transformation of orthogroup- and synteny constrained blast hits. The pan-genome order and

chromosome ID were taken hierarchically, where each orthogroup was positioned by the most likely syntenic position against the 'Pawnee' genome. In the case of orthogroups with a single-copy gene in 'Pawnee', the pan-genome location was simply the rank-order location of that gene in the 'Pawnee' annotation. For orthogroups with multiple members within a genome, the inferred pan-genome position was taken as the location of the most central gene, calculated as the gene with the highest summed blast bit score across the within-genome blast hits. Ties were broken by physical centrality (closest to the median position of the orthogroup) then gene length. For orthogroups without a representative in 'Pawnee', the mean syntenic position of the representative member of each genome was taken as the initial position. Molecular evolution statistics ($K_a$, $K_s$) were calculated from multiple CDS MAFFT v7.470[77] alignments for each single-copy orthogroup in the pan-genome and subsequent analysis in Seqinr 4.2-5[78].

To define candidate variants between haplotypes within each genome, we projected the closest representative of each pan-genome orthogroup against the alternative haplotype assembly of each genome using gmap v2020-06-30[79]. Protein blast databases between each primary and alternative haplotype annotation were parsed to find the midpoint syntenic location of each alternative haplotype contig. Protein sequences were aligned for each orthologous sequence pair and the percent identity was calculated as $100 \times$ (identical positions) / (aligned positions + internal gap positions).

It is important to note that, while necessary to compare orthologous sequences within user-defined coordinates among genomes, constraining to synteny may induce a slight reduction in precision of all genome-wide orthogroups. This is because small translocations (<min block size) will not be captured as syntenic regions. We checked this by extracting all 1:1 reciprocal best scoring diamond[74] hits (RBHs) from the blast-like database. Overall, we observed 131,869 unique pairwise RBHs. Of these 131,025 were in the syntenic network, revealing very little loss of precision when constraining to synteny.

Previous comparisons of sequences underlying annotation-based presence–absence variation[32] have found that complete sequence deletions rarely underlie regions that lack a gene model (absences) in PAV orthogroups. More commonly, syntenic absences contain similar or nearly identical sequences that did not satisfy the criteria for calling a gene model. In some cases, these are 'low-confidence' genes that barely passed a threshold in the first place. Alternatively, mutations in introns, splice sites or other key positions can reduce evidence for a gene model below a threshold even if the coding sequence is identical. To test for these various patterns of gene absences, we extracted the longest CDS among genes present in an orthogroup and aligned that sequence against the assembly of the genomes containing syntenic absences with gmap[79], allowing only a single best match and outputting a psl-formatted text file. The psl file was parsed to only alignments on the syntenic chromosome of the orthogroup and percent identity (# mismatches / (#mismatches + #matches) and percent coverage ((#matches / CDS length) × 100) were calculated. The resulting alignments were categorized as 'very similar' (>99% sequence coverage, ≥95% sequence identity) 'diverged' (75–99% sequence coverage, 75–95% sequence identity), or 'absent' (0–75% sequence coverage or 0–75% sequence identity). Gene counts are summarized in Supplementary Table 2.

**Genomic introgressions**. A total of 30 DNA samples, extracted using the Qiagen DNAeasy Plant kit (Qiagen, Inc., Valencia, CA), were resequenced at a median depth of 55× (range 38×–214×, Supplementary Table 7), encompassing the four reference genomes, 13 of their relatives, five 'true pecan' genotypes that were known to have little or no interspecific admixture, and eight outgroup samples (*C. cordiformis* = 2, *C. aquatica* = 3, *C. myristiciformis* = 3). The samples were sequenced using Illumina HiSeq paired-end sequencing (2 × 150 bp) at the HudsonAlpha Institute for Biotechnology (Huntsville, AL). The reads were mapped to the 'Oaxaca' assembly using bwa-mem v0.7.12[80]. The resulting bam file was filtered for duplicates using Picard v2.19.0 (http://broadinstitute.github.io/picard). Multi-sample SNP calling was accomplished with SAMtools v1.9[81] mpileup (-Q 20 -d 500) and Varscan v2.4.3[82] with a minimum coverage of 8 and a minimum alternate allele count of 4.

To infer the position and identity of genomic introgressions, we pruned the SNP dataset to sites with a minor allele count of ≥3, no missing data, and maximum linkage disequilibrium $r^2 \leq 0.999$ within 100-SNP windows via bcftools v1.9[83]. The pruned vcf was transformed into reference allele counts (0/1/2). Proportion of ancestry (P$_0$: *C. aquatica* = 0.021, P$_1$: *C. cordiformis* = 0.014, P$_2$: *C. myristiciformis* = 0.075, P$_3$: pecan = 0.890) was inferred with SNPRelate[84].

To infer positions and ancestry of introgression regions, we ran Ancestry_HMM v0.94[85], which leverages allele frequencies in putative parental populations to determine regions of likely introgressions in a test population via a hidden Markov model. For the Ancestry_HMM run, we assumed a recent history of introgression and subsequent backcrossing to true pecan (-p 0 5 1 -p 1 5 1 -p 2 5 1 -p 3 5 1 -p 3 4 .5 -p 3 3 0.25 -p 3 2 0.125 --ne 1000 --tmax 5 -e 1e-3 --tolerance .01 -g) where population 3 (-p 3) is the true pecan and the three potential introgressing species are populations 0–2. Posterior probabilities were converted into hard calls of the most likely genotype, and genotype blocks were calculated by iteratively culling runs of identical calls from two- to 500-marker blocks.

**Differential expression to scab inoculation**. The commercial pecan cultivar, 'Desirable' was used for scab fungal inoculation experiments. Thirty grafted 1-year-old potted trees were split into two groups (15 trees in each): the control group was mock-inoculated with sterilized diH$_2$O while the other group was sprayed until run off with a conidial suspension of scab isolate De-Tif-11 ($1 \times 10^6$ conidia/mL). Trees were placed in a humidity room (cooler with power off, overhead light, and several humidifiers running, 24–27 °C) to maintain free moisture on leaf surfaces for 48 h. Trees were removed and placed in a warehouse with diffuse overhead light provided by interspersed clear ceiling panels (12 h day length, ambient humidity, 20–29 °C) for the remainder of the experiment. Both control and treatment groups were divided into 3 subgroups of 5 trees to serve as replicates. At 24 h post inoculation, 2 leaflets from each tree were collected and frozen with liquid nitrogen. Thus, for each group, there were 3 replicates each containing 10 leaflets (2 each from 5 seedlings). The 24 h time point was chosen to both control for diurnal/circadian gene-expression regulatory patterns and capture the early molecular responses to the presence of the fungus that may be critical in understanding host susceptibility.

Total RNA was isolated and purified from the leaf tissues using the Norgen Plant/Fungi Total RNA Purification Kit (Norgen Biotek Corp., Tharold, Ontario, Canada). 150-bp paired-end sequencing was performed using Illumina HiSeq platform (Illumina, San Diego, CA). Raw reads were checked for quality with FastQC v0.11.8 (http://www.bioinformatics.babraham.ac.uk/projects/fastqc/), adapter trimmed, and filtered for quality and length with Trimmomatic v0.36[86] with default parameters. Processed reads were aligned against the SILVA rRNA database for eukaryotes using Bowtie2 v2.3.4.1[87] to remove any rRNA reads present. Unaligned paired reads were recovered and aligned against the 'Pawnee' reference genome via STAR v2.7[88] with default parameters. Read counts per gene were obtained using HTseq v0.9.1[89]. Linear differential gene-expression analysis was performed via Wald contrasts with DESeq2 v1.28.1[90]. Differentially expressed genes were defined as those with Benjamini-Hochberg adjusted contrast P-value ≤ 0.05 and |log$_2$ fold-change| ≥ 1.5 (Supplementary Data 3). Differentially expressed genes were subjected to gene ontology enrichment analysis using Fisher's exact test in topGO v2.40.0[91]. GO terms were considered significant with Fisher's exact test of <0.05 (Supplementary Table 6).

**'Lakota' × 'Oaxaca' mapping population creation and phenotyping**. Controlled cross progeny were generated by multiple teams using pollen collected from the 'Oaxaca' ortet in Byron, GA, and applying it to receptive flowers on multiple cloned accessions of 'Lakota' during the spring of 2016 and 2017. Progeny nuts were assigned individual numbers, measured, stratified, and planted in pots in a Brownwood TX, greenhouse in March of 2018. In June of 2018, after diameter and height measurements, the progenies were sampled for DNA analysis and randomized in racks in a pecan scab screening nursery at the NCGR-*Carya* in Somerville, TX stratified by an orchard of origin. Seedlings were transplanted in March, 2019, into nursery rows that maintained their randomized positions.

Progeny were monitored for various traits including gall incidence in 2019. The species of *Phylloxera* observed was determined from photographs of the galls compared to verified specimens (Michele R. Warmund, personal communication). While small numbers of pecan leaf phylloxera galls (*Phylloxera notabilis*) were observed, the vast majority of galls had morphologies indicating southern pecan leaf phylloxera (*Phylloxera russellae*). Given some ambiguities in the systematics of the gall-forming pests, we have opted to refer to the incidence of galls due to aphid-like insects as 'phylloxera' here and elsewhere.

Phylloxera gall incidence was monitored by a single-trained rater from 21 to 24 October 2019 by counting or estimating the number of galls on the most affected leaf (worst phylloxera) and the percent of leaves on the seedling showing any galls (percent phylloxera). The incidence of phylloxera galls in 'Lakota' has not been well characterized to our knowledge. Historical documentation shows an unusually high phylloxera susceptibility in 'Mahan'. However, both 'Mahan' and 'Major' were noted to have progeny with variable levels of phylloxera gall incidence[92,93]. While phylloxera typically only results in cosmetic damage, the presence of such a powerful QTL in the commercially important pedigree of 'Lakota' makes this locus of interest to breeders and researchers interested in understanding and controlling for more economically significant insect pests such as pecan stem phylloxera (*Phylloxera devastatrix*), yellow pecan aphid (*Monelliopsis pecanis*), blackmargined pecan aphid (*Monellia caryella*), and black pecan aphids (*Melanocallis caryaefoliae*).

**Mapping population read mapping and variant detection**. Genomic DNA was isolated using a CTAB based method[94] modified for pecan[48] to extract from approximately 150 mg of tender foliar tissue. Samples were RNAse treated and cleaned using the Zymo Genomic Clean and Concentrator Kit (Zymo Research, Irvine, CA). The successful control of pollination was confirmed by the presence of rare alleles contributed by 'Oaxaca' at SSR loci Ga39[95] and/or Wga242[96] at which 'Oaxaca' is homozygous[23]. In order to generate high-density genetic maps, 143 progeny confirmed at Ga39 and Wga242 were selected for resequencing.

Genetic linkage maps totaling 1,196 cM in length were calculated from 11,491 heterozygous SNP loci segregating in the 'Lakota' genome. Due to the relatively high homozygosity of 'Oaxaca', linkage maps for 'Oaxaca' were not generated. Sequencing reads were mapped to the 'Oaxaca' v1.1 reference sequence and variants detected using the pipeline described in Bentley et al.[48] for calling markers from GBS data with the following modifications; the CLC Genomics Workbench

(Qiagen, Germantown, MD) version 12.0.2 was used for mapping. Paired-end reads were trimmed and processed using *Trim Reads 2.3* with the following parameters: quality trim was set to 0.05 with an ambiguous limit of 2, automatic read-through trimming was used, and the first 10 nucleotides and the last 3 nucleotides were removed. The reference sequence used was 'Oaxaca' v1.1. Read mapping parameters were modified so that the insertion and deletion cost = 6, insertion open cost was 6, insertion extend cost = 1, deletion open cost = 6, deletion extend cost = 1, and minimum read length required to match the reference = 85%. After read mapping and prior to variant detection, the sequencing reads were locally realigned with 3 passes using the CLC function *Local Realignment 1.2*. The variants detected via this pipeline with minor allele frequencies < 0.05, heterozygous call frequencies > 0.8, missing call frequencies > 0.1, or where more than two alleles were observed were not tested as part of the linkage analysis. Additionally, 1% of loci with abnormally high or low read depths were discarded. SNP markers were named based on the sequence and position in the 'Oaxaca' reference sequence. This pipeline was also used to reanalyze the GBS sequencing data from Bentley et al.[48] to call markers in 'Major' and 'Mahan' (the parents of 'Lakota') and determine the origin of the 'Lakota' haplotypes.

Informative SNPs were defined as those where 'Oaxaca' alternative and primary alignments were monomorphic and polymorphisms existed between 'Lakota' primary and alternative alleles. Progeny genotypes were used to phase the informative markers from each chromosome into two clusters following Bentley et al.[49]. After clustering, the historical GBS data of 'Mahan' and 'Major' from Bentley et al.[48] was used to define the marker phases so that at phase one loci the alternate allele was derived from 'Mahan' and at phase to loci the alternate allele was derived from 'Major'. Markers were subset to one 'Lakota' informative testcross marker per phase and 25,000 bp bin prioritizing the markers that demonstrated the greatest agreement with the mean haplotype observed 10 SNPs upstream and downstream of the position. Visualization and manual curation were used to remove remaining loci that demonstrated clear patterns of disagreement with local patterns of recombination.

**Linkage map calculation QTL mapping**. Linkage maps and marker/trait associations were calculated in R/qtl2 v0.24[97] with the subset markers input as a backcross population. Framework linkage maps were calculated using *est_map* using the Kosambi function with an error.prob of 0.0165 (Supplementary Data 5). Kinship between samples was calculated using *calc_kinship* and the leave-one-chromosome-out (LOCO) method. Trait associations were calculated using *scan1* with a step of 0.1 and an LMM model. Tracking meiotic recombination in 'Lakota' was accomplished with 11,489 SNP loci where 'Lakota' was heterozygous and 'Oaxaca' was homozygous ('pseudo-testcross' loci; Supplementary Data 5). QTL Bayesian credible (95%) confidence intervals were calculated in R/qtl v1.47-9 and projected onto the physical position of the Oaxaca genome.

**Candidate gene identification**. To document polymorphisms between the 'Lakota' sequence candidate genes and the other three genomes (Oaxaca, Elliott, Pawnee), we carried over all unique pan-genome gene annotations onto the alternative haplotypes and projected each alternative haplotype contig's physical positions onto each main haplotype reference sequence. We extracted single-nucleotide and structural variants from aligned orthologous sequences. Unalignable genes in the middle of contigs were defined as 'absent' while genes without alternative orthologs in regions that lacked an alternative haplotype contig were assumed to be too homozygous for alternative contig assemblies. Candidate genes were determined to be disease associated by manually evaluating the Uniprot knowledgebase (https://www.uniprot.org/) to determine if an orthologue of the best matching *Juglans regia* or *Arabidopsis thaliana* gene had been described as likely to be related to plant immune response functions. Genes containing PFAM motifs PF00931, PF08263, or PF13855 and/or Panther domain PTHR11017 were identified as possible LRR sequences.

**Reporting summary**. Further information on research design is available in the Nature Research Reporting Summary linked to this article.

## Data availability

Data supporting the findings of this work are available within the paper and its Supplementary Information files. A reporting summary for this article is available as a Supplementary Information file. Genome assembly and annotation have been deposited in GenBank under BioProjects PRJNA680555 ('Oaxaca'), PRJNA680556 ('Pawnee'), PRJNA680557 ('Lakota'), and PRJNA680558 ('Elliott'). Genomes and annotations are also available through phytozome: Pawnee, Elliott, Lakota, and Oaxaca. RNA sequencing reads for annotation and fungus-induced gene expression have been deposited under SRA BioProject PRJNA680537. See Supplementary Tables 1, 4, and 7 as well as Supplementary Data 4 for RNA and DNA resequencing short reads SRA identifiers. Resequencing reads for the 'Lakota' × 'Oaxaca' genetic map were deposited on SRA under BioProject number PRJNA679828. Source data are provided with this paper.

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

## Acknowledgements

The pecan trees used in this study are part of a large genetically diverse collection housed at the National Collection of Genetic Resources, Pecans and Hickories, USDA-ARS Pecan Breeding and Genetics, Somerville, and Brownwood, TX, USA, and a Provenance collection located at the USDA-ARS Southeastern Fruit and Tree Nut Research Laboratory in Byron, Georgia, USA. Contact X. Wang (xinwang.wang@usda.gov) for access to germplasm used in this study. The work was funded by USDA NIFA SCRI-2016-51181-25408. The work conducted by the U.S. Department of Energy Joint Genome Institute is supported by the Office of Science of the U.S. Department of Energy

under Contract No DE-AC02-05CH11231. The work conducted by the U.S. Department of Agriculture Crop Germplasm Research Unit is supported by CRIS projects 3091-21000-039-OOD and 3091-21000-042-OOD. This research was supported by funds from the USDA-ARS project funds to CPM (CRIS project 6054-43440-046-00D) and by an ARS Research Participation Program administered by the Oak Ridge Institute for Science and Education (ORISE) through an interagency agreement between the U.S. Department of Energy (DOE) (DE-SC0014664) and USDA. We wish to thank Michelle R. Warmund for her assistance in identifying the species of *Phylloxera* observed across the 'Lakota' × 'Oaxaca' mapping population.

## Author contributions

J.T.L., N.B.B., G.B., K.C., C.M., M.M., R.H., P.E.K., L.J.G., J.S., and J.J.R. wrote the manuscript with contributions from all authors. J.T.L., N.B.B., G.B., J.W.J., A.S., S.M., C.P., K.C., M.J.M., Y.A., C.B., N.K., C.M., M.S., J.C., S.S., P.E.K., J.G., J.R., and J.S. conducted genome/statistical analyses. J.J.R., J.S., C.M., P.C., L.W., N.B.B., R.S.H., P.E.K., Y.A., L.J.G., C.R., X.W., M.J.M., K.K., M.S., R.H., C.B., and C.P. designed and executed experiments. P.C., L.W., N.B.B., L.J.G., X.W., and K.K. conducted phenotyping and field research. J.J.R., H.S.R., K.C., Y.A., J.W., S.R., L.B., and N.B.B. conducted molecular and lab work.

## Competing interests

The authors declare no competing interests.
