## [Peer Review File · Nature Communications]

REVIEWER COMMENTS

Reviewer #1 (Remarks to the Author):

Review of pecan genome.

Altogether an excellent paper; considerable high-quality data that has been extremely well analyzed, and a very well written paper. A couple of very small points only! Why in Fig. 1A have the authors chosen to "criss-cross applesauce" the chromosome numberings of the different accessions so that they have to display the syntenic connections like braid? Are the numberings based on chromosome size instead of homology? For example, it looks like the larger chr1 of Pawnee is homologous to the smaller chr2 of Lakota. I guess I would have numbered them based on homology, but certainly not a huge point. Just an additional thought of this figure - why not make use of color? Would show syntenic blocks better in panel B, for example, for maize and poplar. In the Methods, I would include a sentence or a few on how Ancestry_HMM works, as I had to read up just a bit not being familiar with the approach. I have a good idea how they identified syntenic blocks, but I think the procedural explanation is rather brief. What was the output at each step for example. Indeed, the Methods section is very terse throughout, and I wonder if the authors might flesh it out a bit more to help users of the work. But these are by no means major issues; perhaps economy of words is usually a good thing, but this isn't a journal with print restrictions.

Reviewer #2 (Remarks to the Author):

Lovell et al. detail de novo and haplotype resolved assemblies of four pecan genomes. Genetic variation within and among individuals is then examined to offer insight into the extent of genomic variation and defense traits are linked to such variation. The genome assembly work has been performed using appropriate tools and the analyses appear to be well conducted. Some details of the assembly process and validation are unclear, and these should be detailed in revision. There are no presented analyses that really assess the extent of remaining split alleles in the assembly, for example using coverage or kmer based approaches.

I would suggest that the authors remove first to the post statements from the paper. I would note that there are several plant pan-genome papers already published, some of which are more genuinely pan-genome rather than multiple genomes in nature. For example, <https://www.nature.com/articles/s42003-019-0474-7> details work in *Populus* and there are papers focused on grape and maize. As such, the concept of performing a pan-genome analysis is not the novel aspect of this work. However, the analyses presented are interesting and insightful and it is certainly an achievement to have reached the point of haplotype resolved assemblies.

It is interesting that so many genes are unique to each genome, with 'Pawnee' having the largest number of unique genes. This was also the genome assembled with HiFi data, so I am curious as to whether the input data for the different assemblies affected the ability to detect some of the genes. Are there similar such examples from other systems where this extent of among-genome variation has been reported within a species?

The authors use the terms "ortholog network" and "ortholog group" seemingly interchangeably. As far as I can recall, I have never heard the term ortholog network before, and would suggest using only ortholog group. This is also reflected by Google search results: 148 for "ortholog network", and 157,000 for "ortholog group".

L41 PacBio CSS – Avoid abbreviations in the abstract

L77 While there will indeed be some degree of gene presence/absence, it is somewhat over-reaching at the current time to suggest that these will represent that basis for a large extent of desired trait selection. I think it is probably more realistic to say alleles rather than genes until further evidence is accumulated.

L88 This is not true for outbreeding tree species. For such tree species past genome assemblies usually represented hybrids of the two haplotypes with genome assemblies produced from diploid sequencing data.

L104 Which markers are referred to here?

Lines 163-165: I find it hard to understand which part of the alignment is being referred to. The most obvious would be on at the end of the second line of alignments, but the sequence is represented in all of the cultivars with just one of the haplotypes lacking it for three of them. An annotation in the figure would be helpful.

Line 168: Should the reference to Fig. 1d be Fig. 1c?

Line 224: Should the reference to Fig. 2c be Fig. 2d?

L252-271 The logic of the experimental design is not clear to me here. The authors aimed to identify candidate genes explaining the variable susceptibility to isolates but then examine DE in response to infection by only a single isolate. Would it not have been more logical to perform inoculation using two isolates for which the host has contrasting levels of resistance to infection and to have observed DE between these two infections? As is it, what is observed is a not-so-surprising activation of generic abiotic stress response mechanisms. Until genetic variation linked to infection severity in any of those genes is identified, this result has limited breeding value.

L299 "but the gene model" which specific gene model does this refer to as 22 gene models are referred to.

L311 "Additional, if proven causative, these candidate variants between haplotypes would provide"

L460 How were telomeres identified and confirmed as properly oriented?

L463 Without further explanation is it not easy to see how heterozygous SNPs are determined to be errors in need of correction. Was the data being aligned to the haplotype-purged assembly if the pre-purged version? In general this is the weakest area of the methods and it would not be possible to reproduce the haplotype purging performed based on the details provided in the current methods text. As this is of high importance to the work detailed I would like to see this improved and clarified.

L464 What are the syntenic markers referred to here?

L471 Give version number as this program is in active development and assemblies change between versions.

L476 Here also, which markers are referred to?

L488 Avoid mixing Latin and common species names. This comment holds throughout the paper.

Supplementary note 1 is not very pleasant to read. Readability would be drastically improved with some simple formatting, but in general the note contains a lot of not particularly useful information.

Figure 2a-b: The x-axis says "Oaxaca Chromosome and physical position (Mbp)", but as far as I can tell, there are no scales for physical position, only chromosome numbers.

Reviewer #3 (Remarks to the Author):

In this manuscript, the authors have constructed de novo diploid genomes of four outbred genotypes spanning the diversity of cultivated pecan, including a PacBio CCS chromosome-scale assembly of both haplotypes of the outbred 'Pawnee' cultivar genome. Comparative analysis and a pan-genome integration revealed substantial and likely adaptive interspecific genomic introgressions, including an over-retained haplotype introgressed from bitternut hickory into pecan breeding pedigrees. The authors then leveraged the pan-genome presence-absence and functional variant database between the two outbred haplotypes of the 'Lakota' genome to identify candidate genes for pathogen resistance. The analyses and resources highlight significant progress towards functional and quantitative genomics in highly diverse and outbred crops. It will be valuable trait-associated markers and other molecular breeding tools for long-lived outbred crop species, such as trees.

However, I still have couple of major concerns as below:

1. The transcriptomes of pecan scab response section: the authors selected the susceptible cultivar "desirable" for defining the gene networks that drive fungal pathogen resistance. Firstly, the transcriptomic data lack qPCR (or other method) validation. Secondly, the authors didn't give the gene networks to drive fungal pathogen resistance in the manuscript, and in my opinion, it is impossible to give these here, because the differentially expressed genes identified here are only "scab response genes" in the susceptible cultivar. The third, the authors didn't use a more strict parameters for identifying the differentially expressed genes (DEGs), it is abnormal that only 194 genes were differentially expressed between control and inoculated treatments. Meanwhile, the categories of GO-enriched DEGs are very normal in many similar studies and it seems far-fetched to draw a conclusion that "these differentially expressed genes offer a set of high-value targets to explore resistance and susceptibility to *V. effusa* and develop disease resistant pecan cultivars.". Nevertheless, it may be necessary to use scab resistant cultivars, such as Excel, Kanza, etc, for control test and comparative analyses to obtain the gene networks.
2. In the abstract of the manuscript, the authors emphasized that "We then leveraged the pan-genome presence-absence and functional variant database between the two outbred haplotypes of the 'Lakota' genome to identify candidate genes for pathogen resistance." However, the authors didn't directly provide evidence for identifying candidate genes for pathogen resistance based on the Lakota genome, instead, the authors speculate the potential use of pathogen resistance QTL identification by phyloxera resistance as example. I wonder why the authors didn't directly use the scab resistance and susceptible mapping populations. Moreover, the authors confused me on the words of pest, insect and pathogen. As we know, the plant use different response mechanisms for pests (maybe including insects) and pathogens, therefore the difference of response gene sets. Moreover, the QTL identified here is too long (>1Mb), to develop it as a screening marker, a lot of experiments are necessary.

Minor concern:

Several mistakes in writing, for example, line 151, should be "pan-genome" instead of "pangenome"; line 199: "Mbp", the authors use "Mb" across the manuscript but this sentence, should be unified.

Reviewer #4 (Remarks to the Author):

The manuscript by Lovell et al describes a first step towards the pangenome exploration of pecan trees, which are long-lived and outcrossing and are thus challenging from a genomics point of view.

1. Novelty

As mentioned by the authors in L94, probably the most interesting part of this study, from a plant genomics perspective, is that they sought to obtain separate haploid genomes (haplotypes) for their heterozygous plants: "have expanded upon these efforts — instead of collapsing two divergent haplotypes into a single haploid assembly, we sought to build diploid assemblies and capture both haplotypes in four outbred pecan genotypes". While it is certainly early days for this approach, as technology did not support it until recently, this has already been done in plants (L115). See for instance <https://www.nature.com/articles/s41588-020-0699-x>

Nevertheless, this approach allows them to estimate the homozygous fraction of the genome and the fraction that requires separate haplotypes (L106), or unprecedented slow rates of chr rearrangements (L127). Their highly contiguous assemblies also support a synteny-based pangenome analysis (L150), which can be used to detect missing blocks of genes (L171), or to carry out the high resolution QTL analysis illustrated on Figure 3, but at the cost of potentially missing orthologues involved in structural variation (L152).

In fact, the authors should estimate how often this occurs by comparing their results to an independent experiment where synteny was not taken into account. This is important as the contiguity of genomes is constantly improving and people out there need to know objectively what is gained and lost when leveraging synteny on a pangenome analysis.

2. Justification of the choice of genotypes.

In L96 and the first section of methods the authors explain the pedigree of the four sequenced genotypes (Oaxaca, Elliott, Lakota and Pawnee), the last two sharing a close ancestor. I have a couple of comments:

2.1) Please provide accession/stock codes for all four genotypes so that readers can order them if required and to ensure traceability from the raw materials to the sequences.

2.2) The authors should present diversity data to show how representative these 4 genotypes are of the pecan germplasm and pangenome. Why was cv Desirable left out? The answers to these questions are relevant to put in the right context their PAV analyses.

3. PAV analysis and private genes

In the paragraphs at L166 and L215, the authors summarize their results of non-core, private and introgressed genes.

After reading the Methods section I understand they produced gene annotation with sufficient supporting RNAseq data, but it is not clear whether they attempted to confirm genes/blocks missing in one genotype by direct read mapping and computing of read depth. If not done I suggest they do that to get quality control estimate.

4. QTL analysis

It's a pity the authors could not single out an individual candidate gene for Phylloxera resistance (L309). The risk of ending their story like that is that sceptics out there might conclude that pangenome analyses are not really worth the effort and cost, as similar results could have been obtained with GBS markers. Perhaps they could compensate by comparing their analysis to a standard GBS/SNP marker analysis to measure the gain in performance and resolution.

5. Data availability

While the sequence reads have been submitted to the NCBI GenBank, the assembly and annotation is only available from phytozome, which is not a INSDC public archive. I strongly recommend these assemblies are submitted to GenBank as well, at the very least cultivar Pawnee, for two reasons:

- 5.1) For long-term sustainability of these assemblies, which will be assigned unique accessions
- 5.2) In my experience, common errors (contamination, assembly errors) are only spotted during submission of the assembly.

6. Minor issues:

I invite the authors to use the terms 'core', 'shell/accessory' and 'cloud' which have been used in the literature for a while now.

Below, we have copied the reviews and supplied direct responses to each comment. Where appropriate, our responses include the line numbers in the new manuscript version where changes have been made.

Reviewer #1:

Altogether an excellent paper; considerable high-quality data that has been extremely well analyzed, and a very well written paper. A couple of very small points only!

Why in Fig. 1A have the authors chosen to “criss-cross applesauce” the chromosome numberings of the different accessions so that they have to display the syntenic connections like braid? Are the numberings based on chromosome size instead of homology? For example, it looks like the larger chr1 of Pawnee is homologous to the smaller chr2 of Lakota. I guess I would have numbered them based on homology, but certainly not a huge point. Just a an additional thought of this figure - why not make use of color? Would show syntenic blocks better in panel B, for example, for maize and poplar.

- **Thanks for these suggestions. The chromosomes are labeled by homology, and the braiding is caused by homeologous adjacent chromosomes. We have made several improvements to figure 1 to help improve clarity, especially coloring the homeologous and orthologous chromosomal regions differently.**

In the Methods, I would include a sentence or a few on how Ancestry_HMM works, as I had to read up just a bit not being familiar with the approach.

- **We have expanded the description and utility of Ancestry_HMM [574-576].**

I have a good idea how they identified syntenic blocks, but I think the procedural explanation is rather brief. What was the output at each step for example.

- **We have expanded the discussion of the GENESPACE pipeline [530-534].**

Indeed, the Methods section is very terse throughout, and I wonder if the authors might flesh it out a bit more to help users of the work. But these are by no means major issues; perhaps economy of words is usually a good thing, but this isn't a journal with print restrictions.

- **We had initially attempted to streamline the methods section, but we agree with this comment and have expanded the methods accordingly.**

Reviewer #2:

Lovell et al. detail de novo and haplotype resolves assemblies of four pecan genomes. Genetic variation within and among individuals is then examined to offer insight into the extend of genomic variation and defense traits are linked to such variation. The genome assembly work has been performed using appropriate tools and the analyses appear to be well conducted. Some details of the assembly process and validation are unclear, and these should be detailed in revision. There are no presented analyses that really assess the extent of remaining split alleles in the assembly, for example using coverage or kmer based approaches.

I would suggest that the authors remove first to the post statements from the paper. I would note that there are several plant pan-genome papers already published, some of which are more genuinely pan-genome rather than multiple genomes in nature. For example, <https://www.nature.com/articles/s42003-019-0474-7> details work in *Populus* and there are papers focused on grape and maize. As such, the concept of performing a pan-genome analysis is not the novel aspect of this work. However, the analyses presented are interesting and insightful and it is certainly an achievement to have reached the point of haplotype resolved assemblies.

- **Thanks for this suggestion. We agree that novelty can be a difficult thing to state in a paper and have dropped the references to novelty. We will say though, that nearly all the work in poplar, maize and other systems use short read draft pan-genome annotations (including the linked article), which offer far less precision than we have here. All that said, there have been some recent chromosome-scale pangenomes in inbred systems like tomato and soybean, so your point is well taken.**

It is interesting that so many genes are unique to each genome, with ‘Pawnee’ having the largest number of unique genes. This was also the genome assembled with HiFi data, so I am curious as to whether the input data for the different assemblies affected the ability to detect some of the genes. Are there similar such examples from other systems where this extent of among-genome variation has been reported within a species?

- **This is an interesting point. The genic regions of all four assemblies are very complete. The primary increase in contiguity in HiFi/CCS from CLR comes from repeat-rich or variable heterozygosity regions. The latter pattern may be more of an indicator since there is extensive PAV between the haploid genomes. The level of annotation evidence was similar between the genomes, so this is not likely to be the cause.**
- **It’s worth noting that this number of unique genes really isn’t that high for substantially diverged genomes. For example, in our *Panicum hallii* subspecies (~1M y diverged, so quite a bit farther apart than the pecan genomes), we observed 9500 unique genes in just one of the two de novo assemblies/annotations (<https://www.nature.com/articles/s41467-018-07669-x/tables/1>).**

The authors use the terms “ortholog network” and “ortholog group” seemingly interchangeably. As far as I can recall, I have never heard the term ortholog network before, and would suggest using only ortholog group. This is also reflected by Google search results: 148 for “ortholog network”, and 157,000 for “ortholog group”.

- **We have changed the term ‘orthology network’ to ‘orthogroup’ to be consistent throughout.**

L41 PacBio CSS – Avoid abbreviations in the abstract

- **Changed to ‘circular consensus sequence’ ... although, we think more researchers know it by HiFi, which is a trade name we’d prefer to avoid.**

L77 While there will indeed be some degree of gene presence/absence, it is somewhat over-reaching at the current time to suggest that these will represent that basis for a large extent of desired trait selection. I think it is probably more realistic to say alleles rather than genes until further evidence is accumulated.

- **This point is well taken. We have hedged slightly in our assertion.**

L88 This is not true for outbreeding tree species. For such tree species past genome assemblies usually represented hybrids of the two haplotypes with genome assemblies produced from diploid sequencing data.

- **We have altered the text here to point out that we are using natural outcrossing genotypes and not F1s.**

L104 Which markers are referred to here?

- **The markers are detailed in the methods, but, since describing this effort in the main text would be tedious for the reader, we have opted to re-frame this section a bit to make it read more clearly.**

Lines 163-165: I find it hard to understand which part of the alignment is being referred to. The most obvious would be on at the end of the second line of alignments, but the sequence is represented in all of the cultivars with just one of the haplotypes lacking it for three of them. An annotation in the figure would be helpful.

- **Agreed. We have clarified the description.**

Line 168: Should the reference to Fig. 1d be Fig. 1c?

- **Upon revisiting, yes, 1C would be more appropriate. We have altered this accordingly.**

Line 224: Should the reference to Fig. 2c be Fig. 2d?

- **Yes, it should. Changed.**

L252-271 The logic of the experimental design is not clear to me here. The authors aimed to identify candidate genes explaining the variable susceptibility to isolates but then examine DE in response to infection by only a single isolate. Would it not have been more logical to perform inoculation using two isolates for which the host has contrasting levels of resistance to infection and to have observed DE between these two infections? As is it, what is observed is a not-so-surprising activation of generic abiotic stress response mechanisms. Until genetic variation linked to infection severity in any of those genes is identified, this result has limited breeding value.

- **This is a good point. We should have framed this analysis, less as a search for candidate genes for functional variation among pecan, which as you point out cannot be tested by this experimental design, and more directly as a test of phenotypic plasticity in a susceptible variety. Since most pecan cultivars are susceptible to scab and (as we now note) Desirable is becoming increasingly susceptible, we felt that this approach would be more broadly useful than a targeted differential expression analysis between pecan cultivars. We have altered the text accordingly [253-254 and elsewhere in this section]. This said, we do plan to conduct an analysis similar to what is proposed here in the near future.**

L299 “but the gene model” which specific gene model does this refer to as 22 gene models are referred to.

- **We have altered the text to better describe this situation**

L311 “Additional, if proven causative, these candidate variants between haplotypes would provide”

- **We have altered the text accordingly**

L460 How were telomeres identified and confirmed as properly oriented?

- **We have added text describing this method, including the specific kmer [498-500].**

L463 Without further explanation is it not easy to see how heterozygous SNPs are determined to be errors in need of correction. Was the data being aligned to the haplotype-purged assembly if the pre-purged version? In general this is the weakest area of the methods and it would not be possible to reproduce the haplotype purging performed based on the details provided in the current methods text. As this is of high importance to the work detailed I would like to see this improved and clarified.

- **This is standard polishing protocol, where heterozygous SNPs are issues with phasing not sequencing errors per se. We now state this [469].**

L464 What are the syntenic markers referred to here?

- **We now provide more detail about how these were generated to do syntenic builds [473-475, 485-487].**

L471 Give version number as this program is in active development and assemblies change between versions.

- **Good point. Added.**

L476 Here also, which markers are referred to?

- **See above.**

L488 Avoid mixing Latin and common species names. This comment holds throughout the paper.

- **Fixed throughout.**

Supplementary note 1 is not very pleasant to read. Readability would be drastically improved with some simple formatting, but in general the note contains a lot of not particularly useful information.

- **We were on the fence about including this note at all. We have decided to drop it and include the details in the methods instead.**

Figure 2a-b: The x-axis says “Oaxaca Chromosome and physical position (Mbp)“, but as far as I can tell, there are no scales for physical position, only chromosome numbers.

- **Good point. The axis is scaled by physical position and labeled by chromosome. We have added a specific scale bar and modified the axis and caption accordingly.**

Reviewer #3:

In this manuscript, the authors have constructed de novo diploid genomes of four outbred genotypes spanning the diversity of cultivated pecan, including a PacBio CCS chromosome-scale assembly of both haplotypes of the outbred ‘Pawnee’ cultivar genome. Comparative analysis and a pan-genome integration revealed substantial and likely adaptive interspecific genomic introgressions, including an over-retained haplotype introgressed from bitternut hickory into pecan breeding pedigrees. The authors then leveraged the pan-genome presence-absence and functional variant database between the two outbred haplotypes of the ‘Lakota’ genome to identify candidate genes for pathogen resistance. The analyses and resources highlight significant progress towards functional and quantitative genomics in highly diverse and outbred crops. It will be valuable trait-associated markers and other molecular breeding tools for long-lived outbred crop species, such as trees. However, I still have couple of major concerns as below:

1. The transcriptomes of pecan scab response section: the authors selected the susceptible cultivar “desirable” for defining the gene networks that drive fungal pathogen resistance. Firstly, the transcriptomic data lack qPCR (or other method) validation.

- **We believe the question is why we have not looked for congruence between PCR-based and RNA-seq transcript abundance assays. If this is not the correct interpretation, then please ignore the response below and clarify in future correspondence.**
- **We do not believe that checking for qPCR-RNAseq concordance would be useful for the scab RNA-seq analysis and have not included these results for three reasons:**
 - a. **There is substantial literature showing that qPCR validation is not necessary for RNA-seq based transcript abundance assays. While not a novel discovery (Hughes, 2009, 10.1186/jbiol104), the full transition of the genomics community to RNA-seq technology away from microarrays has led to a large number of comparative analyses showing little evidence for a need to ‘validate’ RNA-seq results (Everaert et al. 2017,10.1038/s41598-017-01617-3; Coenye 2021, 10.1016/j.bioflm.2021.100043; many others). To this point, most recent global assays of gene expression via RNA sequencing do not undertake qPCR ‘validation’.**
 - b. **The issue of minor discordance between different methods is driven mostly by genes that are not all that differentially expressed (Everaert et al. 2017). Global analyses of only highly differentially expressed genes (as we do here) likely gets around this problem.**
 - c. **We agree that checking for concordance among methods remains an important tool to explore expression of small sets of specific candidate genes. However, such validation is not necessary in exploratory analyses where the goal is to understand the expression landscape (as is our goal) and not to define functional variants.**

Secondly, the authors didn’t give the gene networks to drive fungal pathogen resistance in the manuscript, and in my opinion, it is impossible to give these here, because the differentially expressed genes identified here are only “scab response genes” in the susceptible cultivar.

- **Agreed. Perhaps this title of this section was a bit overzealous. We have altered this and other interpretation to be more in line with the scale of inference possible with our experimental design. See our response to reviewer 2 above. In short, we have now framed**

this section as assessment of plasticity to a biotic factor that most pecan cultivars are susceptible to.

The third, the authors didn't use a more strict parameters for identifying the differentially expressed genes (DEGs), it is abnormal that only 194 genes were differentially expressed between control and inoculated treatments.

- **We are not sure we follow this comment. The number of genes that are differentially expressed is a function of the biology of the system and the statistical tests. We feel 194 DE genes represents a reasonable number of genes for this experiment.**

Meanwhile, the categories of GO-enriched DEGs are very normal in many similar studies and it seems far-fetched to draw a conclusion that “these differentially expressed genes offer a set of high-value targets to explore resistance and susceptibility to *V. effusa* and develop disease resistant pecan cultivars.”.

Nevertheless, it may be necessary to use scab resistant cultivars, such as Excel, Kanza, etc, for control test and comparative analyses to obtain the gene networks.

- **As described above, we have altered the interpretation to be more in line with the study of plasticity and less about finding resistance loci. However, we disagree with the assertion that understanding phenotypic plasticity of a susceptible cultivar is not useful for probing the genetic basis of plant responses to biotic interactions. While it is true that having more resistant cultivars in the experimental design would have improved inference of genotype-by-environment interactions that may underlie cultivar-specific resistance, this was not the goal of this study.**

2. In the abstract of the manuscript, the authors emphasized that “We then leveraged the pan-genome presence-absence and functional variant database between the two outbred haplotypes of the ‘Lakota’ genome to identify candidate genes for pathogen resistance.” However, the authors didn't directly provide evidence for identifying candidate genes for pathogen resistance based on the Lakota genome, instead, the authors speculate the potential use of pathogen resistance QTL identification by phylloxera resistance as example.

- **We have changed the word pathogen to pest. We think this is the issue you are describing? If not, then please note that we did produce a database of putatively functional and PAV variants for the phylloxera QTL in the supplement (s table 4) and describe those in the main text.**

I wonder why the authors didn't directly use the scab resistance and susceptible mapping populations.

- **This is not in the scope of this study.**

Moreover, the authors confused me on the words of pest, insect and pathogen. As we know, the plant use different response mechanisms for pests (maybe including insects) and pathogens, therefore the difference of response gene sets.

- **We have altered the text accordingly**

Moreover, the QTL identified here is too long (>1Mb), to develop it as a screening marker, a lot of experiments are necessary.

- **It certainly is true that finding the causal nucleotides underlying a linkage mapping QTL will require follow up experiments. That's linkage mapping, by definition: the cost in precision of linkage mapping is balanced by accurate and causal inference.**
- **This locus can be targeted directly for improvement without knowing the causal variant(s) (i.e. the causal QTN is not needed for marker assisted selection). Indeed, we do not follow how a QTL region can be 'too large' to develop screening markers. Markers are chosen at the peak position, regardless of the interval size.**
- **All this said, we feel that a 1Mb interval with <50 genes offers an excellent system to find candidate genes. Further, 1Mb is a very reasonable QTL size for an F1 genetic map in Pecan. Perhaps in other systems with smaller genomes or more recombination, this would be too large for some practical uses, but not here.**

Minor concern: Several mistakes in writing, for example, line 151, should be "pan-genome" instead of "pangenome"; line 199: "Mbp", the authors use "Mb" across the manuscript but this sentence, should be unified.

- **These have been altered accordingly.**

Reviewer #4:

The manuscript by Lovell et al describes a first step towards the pangenome exploration of pecan trees, which are long-lived and outcrossing and are thus challenging from a genomics point of view.

As mentioned by the authors in L94, probably the most interesting part of this study, from a plant genomics perspective, is that they sought to obtain separate haploid genomes (haplotypes) for their heterozygous plants: "have expanded upon these efforts — instead of collapsing two divergent haplotypes into a single haploid assembly, we sought to build diploid assemblies and capture both haplotypes in four outbred pecan genotypes". While it is certainly early days for this approach, as technology did not support it until recently, this has already been done in plants (L115). See for instance <https://www.nature.com/articles/s41588-020-0699-x> Nevertheless, this approach allows them to estimate the homozygous fraction of the genome and the fraction that requires separate haplotypes (L106), or unprecedented slow rates of chr rearrangements (L127). Their highly contiguous assemblies also support a synteny-based pangenome analysis (L150), which can be used to detect missing blocks of genes (L171), or to carry out the high resolution QTL analysis illustrated on Figure 3, but at the cost of potentially missing orthologues involved in structural variation (L152).

In fact, the authors should estimate how often this occurs by comparing their results to an independent experiment where synteny was not taken into account. This is important as the contiguity of genomes is constantly improving and people out there need to know objectively what is gained and lost when leveraging synteny on a pangenome analysis.

- **This is a good idea and one that our pipeline does automatically. We have added a section to this end in the methods [554-559] and now provide the number of 1:1 reciprocal best diamond (blast-like) hits that are syntenic (131,025) and non-syntenic (844).**

In L96 and the first section of methods the authors explain the pedigree of the four sequenced genotypes (Oaxaca, Elliott, Lakota and Pawnee), the last two sharing a close ancestors. I have a couple of comments:

2.1) Please provide accession/stock codes for all four genotypes so that readers can order them if required and to ensure traceability from the raw materials to the sequences.

- **Done.**

2.2) The authors should present diversity data to show how representative these 4 genotypes are of the pecan germplasm and pangenome. Why was cv Desirable left out? The answers to these questions are relevant to put in the right context their PAV analyses.

- **We would love to present a full pop gen analysis placing the four references in context of others, but this data is not ready yet. We hope to have the quantitative/population genomics data compiled later in 2021, but this is not within the scope of the present paper (we only present a small proportion of the full diversity data).**
- **As for Desirable, it was left out because we wanted one genotype from a non-breeding pedigree (Oaxaca) and had to make the tough decision.**

In the paragraphs at L166 and L215, the authors summarize their results of non-core, private and introgressed genes. After reading the Methods section I understand they produced gene annotation with sufficient supporting RNAseq data, but it is not clear whether they attempted to confirm genes/blocks missing in one genotype by direct read mapping and computing of read depth. If not done I suggest they do that to get quality control estimate.

- **We are not exactly sure what the question here is. Is it whether using homology support from each genome (e.g. similarity of RNA-seq reads that map to alternative references) would improve the annotations and estimates of PAV? If so, see below. If not, please clarify in future correspondence.**
- **First off, we work very hard in our annotations to ensure that we have similar evidence support for each genome annotation. For example, RNA-seq support provided by >187M paired end reads for each genotype. Plus, RNA-seq evidence is only one part of the pipeline that also takes into account homology and a variety of gene model attributes that provide higher or lower confidence scores for a given model. We now detail these efforts more precisely in the methods [496-500].**
- **The homology database that was used for all genomes is identical, so, when we look for sequence similarity in one genome, we do the same for all others. We think this approach is similar to, albeit more stringent (due to whole genes instead of short reads) than what is proposed using RNA-seq directly in the reviewer comment above.**
- **We believe that this combination of homology and RNA-seq evidence gives us gene models that are very robust (see response to reviewer 2 above).**

It's a pity the authors could not single out an individual candidate gene for Phylloxera resistance (L309). The risk of ending their story like that is that sceptics out there might conclude that pangenome analyses are not really worth the effort and cost, as similar results could have been obtained with GBS markers. Perhaps they could compensate by comparing their analysis to a standard GBS/SNP marker analysis to measure the gain in performance and resolution.

- **We respectfully disagree. We cannot envision a way that standard GBS/SNP based approaches would be able to document any functional variation in a candidate region that has substantial PAV in an F1 mapping population. In the case of our candidate regions, reads simply wouldn't map to the missing haplotype in the primary reference assembly.**
- **While the pangenome does not necessarily decrease the size of the candidate gene list (as the peak width is limited by the number of crossover events near the causal variant), it does improve our ability to characterize complex patterns of variation such as PAV. Even a high quality haploid reference would have been incapable of facilitating the type of analysis conducted here.**
- **We have altered the text to highlight the power of this approach and the substantial advances that this offers relative to markers (GBS or otherwise) mapped to a single reference [320-324].**

While the sequence reads have been submitted to the NCBI GenBank, the assembly and annotation is only available from phytozome, which is not a INSDC public archive. I strongly recommend these assemblies are submitted to GenBank as well, at the very least cultivar Pawnee, for two reasons: 5.1) For long-term sustainability of these assemblies, which will be assigned unique accessions 5.2) In my experience, common errors (contamination, assembly errors) are only spotted during submission of the assembly.

- **All genomes and annotations were placed on GenBank and fully vetted. See the data accessibility section.**

I invite the authors to use the terms 'core', 'shell/accessory' and 'cloud' which have been used in the literature for a while now.

- **Where appropriate, we have added these terms.**

REVIEWER COMMENTS

Reviewer #1 (Remarks to the Author):

The authors have now addressed my points satisfactorily.

Reviewer #2 (Remarks to the Author):

The authors have addressed all of my initial comments and I thank them for their comprehensive answers and careful consideration of those comments. I have no further points to raise.

Reviewer #3 (Remarks to the Author):

In the revised manuscript, the authors have responded and revised most of my concerns well. But for the revised version, I still has several concerns. As below,

1. For the title, I noted that the authors changed "A chromosome-scale pan-genome" to "Four chromosome-scale genomes". But the word "pan-genome" is still throughout the manuscript. I wonder whether the change is just to change a more appropriate title.

2. For the section of transcriptomes, my question is still in the experimental design – maybe I didn't make it clear last time: the authors interpreted that "we have now framed this section as assessment of plasticity to a biotic factor that most pecan cultivars are susceptible to" and "and more directly as a test of phenotypic plasticity in a susceptible variety" in their responses to reviewers. The authors have also altered the text in the section and addressed that "these differentially expressed genes (Supplementary Table 267 3) offer a set of targets to explore the processes leading to the host susceptibility to *V. effusa*". Firstly, the authors only used two time points i.e. uninfected control and 24-hour infected samples for the transcriptomic analysis, the differentially expressed gene set are hard to explore "the processes". So, I suggest to use at least three time points (including the control) to explore "the processes". Secondly, in my knowledge, chitin is the main components of fungus cell wall which can lead to the plant hosts response as short as in thirty minutes (100 ug/ml). As a test of short-term gene-expression plasticity to *V. effuse*, did the authors tested the 24-hour inoculation is suitable time length for this test? Thirdly, in my experiences, many genes share similar expression patterns in both resistance and susceptible varieties during the responses to pathogen, so, it is difficult to identify the targets the leading to host susceptibility to fungus only by testing susceptible variety under the circumstance of lacking resistance variety as control.

3. L134: the reference 7 omits the published year.

Reviewer #4 (Remarks to the Author):

Hi, thanks for your edits and responses to my previous queries. I'll comment on them:

>This is a good idea and one that our pipeline does automatically. We have added a section to this end in the methods [554-559] and now provide the number of 1:1 reciprocal best diamond (blast-like) hits that are syntenic (131,025) and non-syntenic (844).

Thanks for this, I think this shows in your own words "very little loss of precision when constraining to synteny". Am I right in saying this also shows there's little to be gained from sinteny-based analysis in your context?

>We would love to present a full pop gen analysis placing the four references in context of others, but this data is not ready yet. We hope to have the quantitative/population genomics data compiled later in 2021, but this is not within the scope of the present paper (we only present a small proportion of the full diversity data). -As for Desirable, it was left out because we wanted one genotype from a non-breeding pedigree (Oaxaca) and had to make the tough decision.

I understand those constraints, but can't you at least say something about how these 4 genotypes represent the pecan germplasm? Are their genetic distances similar to those observed among other cultivars/varieties?

> We are not exactly sure what the question here is. Is it whether using homology support from each genome (e.g. similarity of RNA-seq reads that map to alternative references) would improve the annotations and estimates of PAV? If so, see below. If not, please clarify in future correspondence.

Sorry if I wasn't clear. I'll explain what I mean, it's a computational control to rule out PAV due to annotation differences:

- 3.1) You find genes P1, P2, ... Pn which are private to say Oaxaca.
- 3.2) Extract genomic reads covering those loci R1, R2, ... Rn
- 3.3) Map R1, R2, ... Rn to the assemblies were those genes are not annotated
- 3.4) Compute depth of coverage -> you should get numbers << Oaxaca

> We respectfully disagree. We cannot envision a way that standard GBS/SNP based approaches would be able to document any functional variation in a candidate region that has substantial PAV in an F1 mapping population. In the case of our candidate regions, reads simply wouldn't map to the missing haplotype in the primary reference assembly. -While the pangenome does not necessarily decrease the size of the candidate gene list (as the peak width is limited by the number of crossover events near the causal variant), it does improve our ability to characterize complex patterns of variation such as PAV. Even a high quality haploid reference would have been incapable of facilitating the type of analysis conducted here. -We have altered the text to highlight the power of this approach and the substantial advances that this offers relative to markers (GBS or otherwise) mapped to a single reference [320-324].

Sorry if this comment was too bold. I guess what I meant is that developing markers for that interval and target-sequencing that region in the parents would get you a long way as well, right? For that reason I would remove "whole genome assembly" from L321, as there are ways around it. I do feel like Figure 3 is great and really like the deduced haplotypes. However, I feel this part of the analysis could be improved, although not sure whether this manuscript would be the place for it. I am thinking about developing markers for the 1Mb interval, imputing present/absent candidate genes and then do association/correlation looking for the most likely accessory/dispensable loci explaining most of the variation. Does this make sense?

Below, we have copied the reviews and supplied direct responses to each comment. Where appropriate, our responses include the line numbers in the new manuscript version where changes have been made.

Reviewer #1 (Remarks to the Author):

The authors have now addressed my points satisfactorily.

Reviewer #2 (Remarks to the Author):

The authors have addressed all of my initial comments and I thank them for their comprehensive answers and careful consideration of those comments. I have no further points to raise.

Reviewer #3 (Remarks to the Author):

In the revised manuscript, the authors have responded and revised most of my concerns well. But for the revised version, I still has several concerns. As below,

1. For the title, I noted that the authors changed “A chromosome-scale pan-genome” to “Four chromosome-scale genomes”. But the word “pan-genome” is still throughout the manuscript. I wonder whether the change is just to change a more appropriate title.

- **We believe that the new title is more appropriate since the pan-genome, while a valuable dataset to decipher the genetic variation among genomes, is not in and of itself the resource that permits accelerated breeding — the genomes do that. We feel that it is still appropriate to use the pan-genome in the narrative despite that not being part of the title.**

2. For the section of transcriptomes, my question is still in the experimental design – maybe I didn’t make it clear last time: the authors interpreted that “we have now framed this section as assessment of plasticity to a biotic factor that most pecan cultivars are susceptible to” and “and more directly as a test of phenotypic plasticity in a susceptible variety” in their responses to reviewers. The authors have also altered the text in the section and addressed that “these differentially expressed genes (Supplementary Table 267 3) offer a set of targets to explore the processes leading to the host susceptibility to *V. effusa*”.

Firstly, the authors only used two time points i.e. uninfected control and 24-hour infected samples for the transcriptomic analysis, the differentially expressed gene set are hard to explore “the processes”. So, I suggest to use at least three time points (including the control) to explore “the processes”.

- **We believe that the issue at hand is the word 'process'. We agree that two time points are not sufficient to model a curve. We have removed this mention [270]. Adding another timepoint to a completed project is outside the scope of this study.**

Secondly, in my knowledge, chitin is the main components of fungus cell wall which can lead to the plant hosts response as short as in thirty minutes (100 ug/ml). As a test of short-term gene-expression plasticity to *V. effuse*, did the authors tested the 24-hour inoculation is suitable time length for this test?

- **We chose the 24h timepoint due to both experimental constraints and knowledge of the temporal dynamics of fungal infection.**
- **As the reviewer mentioned, fungal infection and resulting molecular responses can be very rapid. For example, Latham and Rushing (1988) showed that 95% of conidia had germinated by 24h post inoculation and had begun to penetrate the cuticle and grow within the leaf. As such, we can assume that plant responses to fungal infection had begun prior to 24h. Further, 24h represents the minimum span of time possible to both test the effects of fungal inoculation and control for the often-massive diurnal/circadian changes in gene expression. We now mention these considerations in the methods [620-622].**

Thirdly, in my experiences, many genes share similar expression patterns in both resistance and susceptible varieties during the responses to pathogen, so, it is difficult to identify the targets the leading to host susceptibility to fungus only by testing susceptible variety under the circumstance of lacking resistance variety as control.

- **Yes, we agree that there will be some overlap between transcriptional responses of susceptible and resistant varieties.**
- **Due to the constraint on sequencing expenditure we had to focus on studying the phenotypic plasticity of gene expression in susceptible genotypes. However, comparative transcriptomics between susceptible and resistant varieties will shed light on molecular underpinnings and pinpoint candidate genes, which will be considered in our future studies.**

3. L134: the reference 7 omits the published year.

- **Corrected**

Reviewer #4 (Remarks to the Author):

Hi, thanks for your edits and responses to my previous queries. I'll comment on them:

>This is a good idea and one that our pipeline does automatically. We have added a section to this end in the methods [554-559] and now provide the number of 1:1 reciprocal best diamond (blast-like) hits that are syntenic (131,025) and non-syntenic (844). Thanks for this, I think this shows in your own words "very little loss of precision when constraining to synteny".

Am I right in saying this also shows there's little to be gained from sinteny-based analysis in your context?

- **Good question ... the answer is both yes and no.**
- **It is true that very few orthogroups are missed when constraining to both homeologous and meiotically homologous regions among genomes. The number we reported (844 vs. 131,025) shows very few orthogroups that contain genes that are neither in syntenic homeologous nor syntenic homologous regions. As such, globally, there isn't a lot to be gained (or lost, depending on your perspective) by constraining to synteny.**

- **However, there are many orthogroups that contain both homeologous and meiotically homologous gene pairs. This is where synteny constraint becomes crucial. If we were to approach the pan-genome or QTL regions ignorant of synteny, we'd find a very different pattern of PAV. For example, truly missing genes may have a similar sequence to homeologs that are in an orthogroup with a candidate gene in a region. This would obscure the true patterns of PAV. We describe this in the text [140-142].**

>We would love to present a full pop gen analysis placing the four references in context of others, but this data is not ready yet. We hope to have the quantitative/population genomics data compiled later in 2021, but this is not within the scope of the present paper (we only present a small proportion of the full diversity data). -As for Desirable, it was left out because we wanted one genotype from a non-breeding pedigree (Oaxaca) and had to make the tough decision. I understand those constraints, but can't you at least say something about how these 4 genotypes represent the pecan germplasm? Are their genetic distances similar to those observed among other cultivars/varieties?

- **We understand now – how representative are the four reference genomes relative to previously surveyed pecan germplasm? How much genetic diversity is spanned by the references and how related are they to each other. Looking back, we should have seen this, sorry.**
- **Since we do not have this data on hand, the best we can do is present the genetic distances from complexity-reduction sequencing of Bentley et al. (2019). We have now added a panel to Ext. Data Fig. 1, more detail in the methods [454-461], and a short mention of the results in the main text [96-98]. In short, this analysis shows that our genomes represent much of the genetic diversity of pecan, and that they are fairly equally (un)related. Hopefully this is what the reviewer is looking for.**

> We are not exactly sure what the question here is. Is it whether using homology support from each genome (e.g. similarity of RNA-seq reads that map to alternative references) would improve the annotations and estimates of PAV? If so, see below. If not, please clarify in future correspondence. Sorry if I wasn't clear. I'll explain what I mean, it's a computational control to rule out PAV due to annotation differences: 3.1) You find genes P1, P2, ... Pn which are private to say Oaxaca. 3.2) Extract genomic reads covering those loci R1, R2, ... Rn 3.3) Map R1, R2, ... Rn to the assemblies where those genes are not annotated 3.4) Compute depth of coverage -> you should get numbers << Oaxaca

- **We thank the reviewer for this clarification.**
- **We conducted a similar analysis in our *Panicum hallii* genome a few years back (Table 1: <https://www.nature.com/articles/s41467-018-07669-x/tables/1>). In that paper, we found that gene-annotation PAV where the sequence is completely deleted tend to be very low quality 'borderline' gene models. But, there are lots of genes where the sequences are very similar but simply do not satisfy the criteria for a functional gene model (e.g. intron structure, splice sites, start/stop variation, presence of an ORF, etc.). While there is a nearly identical sequence between the closely related genotypes, the fact that a gene model cannot be built from one genome's sequence represents true PAV; at least in our view of genome evolution.**
- **We agree that a more nuanced analysis of sequences underlying PAV would be of value for this paper, and we have added an analysis to this effect using full length CDS that should map uniquely. In short, we (1) mapped CDS from PAV orthogroups onto the assemblies of the genomes which were missing members of these subgraphs, (2) extracted alignments of these sequences that were on the expected homologous chromosome, and (3) merged these mappings with our gene model scoring output. Overall, we observe a very similar pattern to *P. hallii*. For example, there are 2798 'high-confidence' PAV gene models that have**

significant homology and expression support in at least one genome. Of these, 28% are completely missing in syntenic regions of genomes and 64% have nearly identical sequences between the genomes with ‘present’ and ‘absent’ genes (the remainder have good mapping but significant sequence variation). The former set represent deletion PAV while the latter represent evidence-based or gene structure PAV. As stated above, we believe both types are true PAV, just different mechanisms of functional ‘presence’. We have added these details to the methods [574-587], built an additional SI table (SI Table 2) and mention some of the results in the main text [168-175].

> We respectfully disagree. We cannot envision a way that standard GBS/SNP based approaches would be able to document any functional variation in a candidate region that has substantial PAV in an F1 mapping population. In the case of our candidate regions, reads simply wouldn't map to the missing haplotype in the primary reference assembly. -While the pangenome does not necessarily decrease the size of the candidate gene list (as the peak width is limited by the number of crossover events near the causal variant), it does improve our ability to characterize complex patterns of variation such as PAV. Even a high quality haploid reference would have been incapable of facilitating the type of analysis conducted here. -We have altered the text to highlight the power of this approach and the substantial advances that this offers relative to markers (GBS or otherwise) mapped to a single reference [320-324]. Sorry if this comment was too bold. I guess what I meant is that developing markers for that interval and target-sequencing that region in the parents would get you a long way as well, right? For that reason I would remove "whole genome assembly" from L321, as there are ways around it. I do feel like Figure 3 is great and really like the deduced haplotypes. However, I feel this part of the analysis could be improved, although not sure whether this manuscript would be the place for it. I am thinking about developing markers for the 1Mb interval, imputing present/absent candidate genes and then do association/correlation looking for the most likely accessory/dispensable loci explaining most of the variation. Does this make sense?

- **We agree that a local assembly would be just as good as a whole genome assembly and have altered line 321 accordingly [now lines 327-329].**
- **Yes, we agree that exploring genetic diversity outside of the F1 population could prove powerful to find targets in this region. We hope to do similar tests once further genetic resources and resequencing are available.**

REVIEWERS' COMMENTS

Reviewer #3 (Remarks to the Author):

I have no further points to raise.

Reviewer #4 (Remarks to the Author):

I have now read the revised version of the manuscript and the responses to my previous questions. Overall I am pleased with the author's efforts and edits. There's only one issue remaining from my part, see 1):

1) "However, there are many orthogroups that contain both homeologous and meiotically homologous gene pairs. This is where synteny constraint becomes crucial. If we were to approach the pan-genome or QTL regions ignorant of synteny, we'd find a very different pattern of PAV. For example, truly missing genes may have a similar sequence to homeologs that are in an orthogroup with a candidate gene in a region. This would obscure the true patterns of PAV. We describe this in the text [140-142]."

Do you mean that without enforcing synteny you get many clusters mixing paralogues? If that's the case, I suggest two changes to the sentence in L140:

1.1) pan-genome construction methods based on traditional sequence amino acid similarity (i.e. BLASTP) are not, on their own, sufficient for comparative genomics since paralogous sequences would likely pollute otherwise orthologous genefamilies.

1.2) Add some numbers from your own analysis to support this, just as you do on L148.

2) "We understand now – how representative are the four reference genomes relative to previously surveyed pecan germplasm? How much genetic diversity is spanned by the references and how related are they to each other. Looking back, we should have seen this, sorry."

That's fine, it's complex story and you have many reviewers :-)

"Since we do not have this data on hand, the best we can do is present the genetic distances from complexity-reduction sequencing of Bentley et al. (2019). We have now added a panel to Ext. Data Fig. 1, more detail in the methods [454-461], and a short mention of the results in the main text [96-98]. In short, this analysis shows that our genomes represent much of the genetic diversity of pecan, and that they are fairly equally (un)related. Hopefully this is what the reviewer is looking for."

This panel helps put the four assemblies in the right context, thanks for that.

3) "We conducted a similar analysis in our *Panicum hallii* genome a few years back (Table 1: <https://www.nature.com/articles/s41467-018-07669-x/tables/1>). In that paper, we found that gene-annotation PAV where the sequence is completely deleted tend to be very low quality 'borderline' gene models. But, there are lots of genes where the sequences are very similar but simply do not satisfy the criteria for a functional gene model (e.g. intron structure, splice sites, start/stop variation, presence of an ORF, etc.). While there is a nearly identical sequence between the closely related genotypes, the fact that a gene model cannot be built from one genome's sequence represents true PAV; at least in our view of genome evolution."

That's reasonable.

"We agree that a more nuanced analysis of sequences underlying PAV would be of value for this paper, and we have added an analysis to this effect using full length CDS that should map uniquely. In short, we (1) mapped CDS from PAV orthogroups onto the assemblies of the genomes which were

missing members of these subgraphs, (2) extracted alignments of these sequences that were on the expected homologous chromosome, and (3) merged these mappings with our gene model scoring output. Overall, we observe a very similar pattern to *P. hallii*. For example, there are 2798 'high-confidence' PAV gene models that have significant homology and expression support in at least one genome. Of these, 28% are completely missing in syntenic regions of genomes and 64% have nearly identical sequences between the genomes with 'present' and 'absent' genes (the remainder have good mapping but significant sequence variation). The former set represent deletion PAV while the latter represent evidence-based or gene structure PAV. As stated above, we believe both types are true PAV, just different mechanisms of functional 'presence'. We have added these details to the methods [574-587], built an additional SI table (SI Table 2) and mention some of the results in the main text [168-175]."

This is a great addition to the manuscript.

4) "We agree that a local assembly would be just as good as a whole genome assembly and have altered line 321 accordingly [now lines 327-329]."

That sentence is perfectly reasonable.

Bruno Contreras Moreira

Below, we have copied the reviews and supplied direct responses to each comment. Where appropriate, our responses include the line numbers in the new manuscript version where changes have been made.

Reviewer #1 (Remarks to the Author):

The authors have now addressed my points satisfactorily.

Reviewer #2 (Remarks to the Author):

The authors have addressed all of my initial comments and I thank them for their comprehensive answers and careful consideration of those comments. I have no further points to raise.

Reviewer #3 (Remarks to the Author):

I have no further points to raise.

Reviewer #4 (Remarks to the Author):

I have now read the revised version of the manuscript and the responses to my previous questions. Overall I am pleased with the author's efforts and edits. There's only one issue remaining from my part, see 1):

1) "However, there are many orthogroups that contain both homeologous and meiotically homologous gene pairs. This is where synteny constraint becomes crucial. If we were to approach the pan-genome or QTL regions ignorant of synteny, we'd find a very different pattern of PAV. For example, truly missing genes may have a similar sequence to homeologs that are in an orthogroup with a candidate gene in a region. This would obscure the true patterns of PAV. We describe this in the text [140-142]."

Do you mean that without enforcing synteny you get many clusters mixing paralogues? If that's the case, I suggest two changes to the sentence in L140:

1.1) pan-genome construction methods based on traditional sequence amino acid similarity (i.e. BLASTP) are not, on their own, sufficient for comparative genomics since paralogous sequences would likely pollute otherwise orthologous gene families.

- **We have altered these two sentences, making the statement regarding BLASTp more general regarding homology as typical pan-genome construction can use global DNA, local DNA or protein alignment methods**

1.2) Add some numbers from your own analysis to support this, just as you do on L148.

- **Done. We added number related to the Pawnee genome homeologs.**

2) "We understand now – how representative are the four reference genomes relative to previously surveyed pecan germplasm? How much genetic diversity is spanned by the references and how related are they to each other. Looking back, we should have seen this, sorry." That's fine, it's complex story and you have many reviewers :-)

"Since we do not have this data on hand, the best we can do is present the genetic distances from complexity-reduction sequencing of Bentley et al. (2019). We have now added a panel to Ext. Data Fig. 1, more detail in the methods [454-461], and a short mention of the results in the main text [96-98]. In short, this analysis shows that our genomes represent much of the genetic diversity of pecan, and that they are fairly equally (un)related. Hopefully this is what the reviewer is looking for." This panel helps put the four assemblies in the right context, thanks for that.

3) "We conducted a similar analysis in our *Panicum hallii* genome a few years back (Table 1: <https://www.nature.com/articles/s41467-018-07669-x/tables/1>). In that paper, we found that gene-annotation PAV where the sequence is completely deleted tend to be very low quality 'borderline' gene models. But, there are lots of genes where the sequences are very similar but simply do not satisfy the criteria for a functional gene model (e.g. intron structure, splice sites, start/stop variation, presence of an ORF, etc.). While there is a nearly identical sequence between the closely related genotypes, the fact that a gene model cannot be built from one genome's sequence represents true PAV; at least in our view of genome evolution." That's reasonable.

"We agree that a more nuanced analysis of sequences underlying PAV would be of value for this paper, and we have added an analysis to this effect using full length CDS that should map uniquely. In short, we (1) mapped CDS from PAV orthogroups onto the assemblies of the genomes which were missing members of these subgraphs, (2) extracted alignments of these sequences that were on the expected homologous chromosome, and (3) merged these mappings with our gene model scoring output. Overall, we observe a very similar pattern to *P. hallii*. For example, there are 2798 'high-confidence' PAV gene models that have significant homology and expression support in at least one genome. Of these, 28% are completely missing in syntenic regions of genomes and 64% have nearly identical sequences between the genomes with 'present' and 'absent' genes (the remainder have good mapping but significant sequence variation). The former set represent deletion PAV while the latter represent evidence-based or gene structure PAV. As stated above, we believe both types are true PAV, just different mechanisms of functional 'presence'. We have added these details to the methods [574-587], built an additional SI table (SI Table 2) and mention some of the results in the main text [168-175]." This is a great addition to the manuscript.

4) "We agree that a local assembly would be just as good as a whole genome assembly and have altered line 321 accordingly [now lines 327-329]." That sentence is perfectly reasonable.